# MTS-UNMixers: Multivariate Time Series Forecasting via Channel-Time Dual Unmixing

## Abstract

Multivariate time series data provide a robust framework for future predictions by leveraging information across multiple dimensions, ensuring broad applicability in practical scenarios. However, their high dimensionality and mixing patterns pose significant challenges in establishing an interpretable and explicit mapping between historical and future series, as well as extracting long-range feature dependencies. To address these challenges, we propose a channel-time dual unmixing network for multivariate time series forecasting (named MTS-UNMixers), which decomposes the entire series into critical bases and coefficients across both the time and channel dimensions. This approach establishes a robust sharing mechanism between historical and future series, enabling accurate representation and enhancing physical interpretability. Specifically, MTS-UNMixers represent sequences over time as a mixture of multiple trends and cycles, with the time-correlated representation coefficients shared across both historical and future time periods. In contrast, sequence over channels can be decomposed into multiple tick-wise bases, which characterize the channel correlations and are shared across the whole series. To estimate the shared time-dependent coefficients, a vanilla Mamba network is employed, leveraging its alignment with directional causality. Correspondlingly, a bidirectional Mamba network is utilized to model the shared channel-correlated bases, accommodating noncausal relationships. Experimental results show that MTS-UNMixers significantly outperform existing methods on multiple benchmark datasets.

## 1 Introduction

The time series forecasting aims to provide accurate predictions of future series values by analyzing time-dependent patterns and trends in historical data. It is a core task in data analytics and is widely used in financial markets Chen et al. (2012), weather forecasting Angryk et al. (2020); Schultz et al. (2021), electric power forecasting Khan et al. (2020); Zhu et al. (2023), and traffic flow estimation Chen et al. (2001); Cirstea et al. (2022); Ma et al. (2014), among other fields. With the exponential growth of data volumes and significant advancements in computational power, predictive models now face two key challenges. The first challenge is modeling complex nonlinear relationships over time to accurately capture essential features within long-term sequences. The second involves extracting interactions from multivariate data to better understand the dynamic variations in time series and identify latent patterns. To address these challenges, models have to effectively integrate data from multiple sources and also have robust prediction capabilities.

Recently, many deep learning models have been proposed to address the challenges of long series dependence and multivariate modeling in multivariate time series data. To capture long-series features more effectively, Chen et al. Chen et al. (2024) introduced adaptive multiscale modeling with temporal dynamic inputs, which leverages both local and global information on the time axis. Wang et al. Wang et al. (2024a) integrated multiscale time series by incorporating micro-seasonal information and macro-trend data, utilizing complementary forecasting capabilities to improve overall performance. In order to extract the interaction information in multivariate time series, Liu et al. Liu et al. (2023) redesigned the Transformer architecture to utilize the attention mechanism to capture the correlation between variables, and at the same time utilized the feed forward neural network to extract the temporal features, which effectively enhances the cross-channel correlation extraction. Meanwhile, Li et al. Li et al. (2023) employed a decomposition module to capture inter-channel rela-

tionships. Its relatively simple structure offers significant efficiency advantages over more complex attention-based mechanisms.

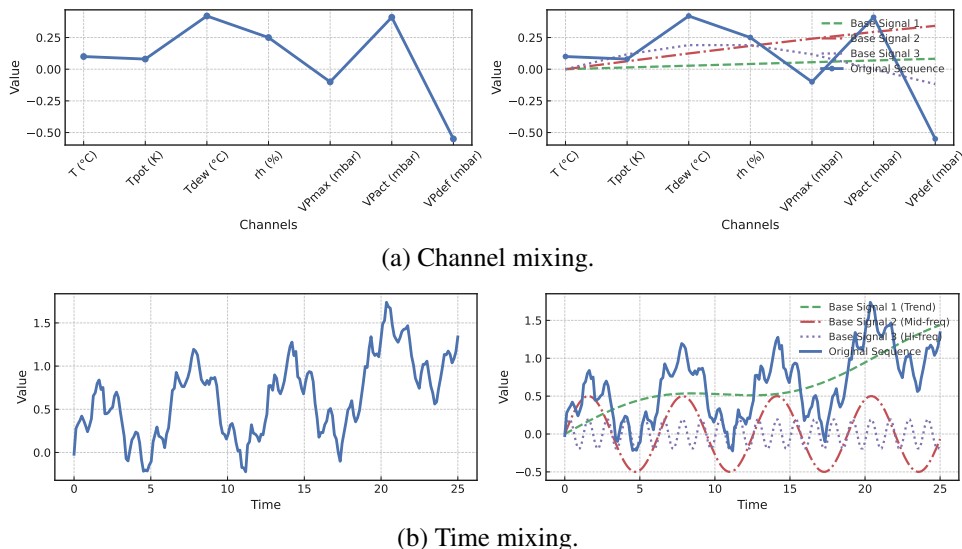

(a) Channel mixing.

(b) Time mixing.

Figure 1: Mixing problem (taking weather data as an example). (a) shows the original plot of the seven variables and their principal component composition in the weather dataset. The high correlation between variables suggests that they may be influenced by common external environmental factors. (b) illustrates the time series of a single channel and shows how it can be decomposed into a mixture of different features.

Despite recent advancements in handling long-term feature series and multivariate interactions, these models still face significant limitations. The first challenge is the problem of mixing in multivariate time series data. Multiple features are intertwined during the modelling process, making it difficult for the model to investigate the individual contribution of each feature. Specifically:

- In the channel dimension, multivariables tend to exhibit high correlation, leading to feature redundancy. As shown in the weather dataset in Fig. 1 (a), variables such as temperature, humidity, and cloud cover are all affected by the dominant temperature trend, which results in similar variations between variables. This similarity reflects the natural correlation between weather variables and may lead to feature overlap and blending, which in turn increases computational complexity and amplifies noise during channel blending.

- In the time dimension, different periods often represent a mixture of patterns, such as cycles and trends, rather than a single feature. For example, in the time series data shown in Fig. 1 (b), we can observe multiple cyclical features (e.g., daily variations or seasonal fluctuations) coexisting with long-term trends. These patterns overlap each other on the time axis, which not only complicates component separation, but also masks localized patterns and interferes with long-term dependencies.

The second challenge lies in the limitations of traditional black-box mapping models in modeling the relationship between historical and future sequences. Black-box mappings typically rely on abstract feature representations and often lack the ability to precisely map between sequences. They also does not have an explicit physical meaning and lacks interpretability of the entire sequence characteristics.

To address the above limitations, we propose MTS-UNMixers, which establishes an explicit mapping of sequences through a unmixing mechanism. MTS-UNMixers decouples the historical sequences in time and channel, and extracts the significant components on the time and channel axes. Specifically, MTS-UNMixers uses a mixture model to represent a single-channel (multi-time) series as a combination of several trend and period components. For a single moment (multi-variate) it

can be represented by a number of cardinalities which describe the correlation between the individual variables, called correlation components.The historical time horizon and the future time horizon are treated as a unified whole, allowing for unmixed and explicit mapping by sharing underlying signal and weight information. In temporal unmixing, we use the Mamba network. The network effectively extracts the nonlinear causal dependencies in the sequence through the dynamic causal mechanism in the state-space model, by utilising the nonlinear multiplication between the basis signal matrix and the component coefficient matrix. In channel unmixing, we employ a bidirectional Mamba network since there is no causal relationship between channels, but rather a bidirectional interaction between highly correlated variables. In this way, the model captures the bidirectional correlations and interactions between variables, realizes the unmixing of independent features in multiple channels, and improves the clarity and accuracy of feature extraction.

In summary, the main contributions of this paper are as follows:

- To address the problem of reduced model accuracy due to feature overlap and correlation in high-dimensional time series data, we propose a dual-mixing model to decompose the entire series into critical bases and coefficients along both the time and channel dimension. This decomposition can effectively investigate the individual contributions of intrinsic time-dependent and channel-correlated bases, reducing the noise and redundancy.

- For effective sharing, we adopt an explicit mapping model. Historical and future sequences are treated as a unified whole, sharing the component coefficient matrix in the channel dimension and the base signal matrix in the time dimension. This improves physical interpretability and prediction reliability while ensuring the conciseness and simplicity of the model.

- In the time dimension, we use Mamba to capture the characteristics of long-term temporal dependencies. In the channel dimension, where no causal relationship exists, we use bidirectional Mamba to effectively capture the bidirectional interactions between variables.

## 2 CHANNEL-TIME MIXING

In this section, we first give a brief introduction to the multivariate time series prediction problem and its bottlenecks. We then formulate the problem. Finally, we describe the process of optimally solving the problem.

### 2.1 PROBLEM STATEMENT

Multivariate Time Series Forecasting (MTSF) is a method that predicts future values by analyzing time series data consisting of multiple variables. Given a historical multivariate time series input $\mathbf{X} \in \mathbb{R}^{T \times N}$, where $T$ represents the number of time steps and $N$ represents the number of variables, the objective is to predict the future target values for the next $H$ time steps. The prediction sequence is denoted as $\hat{\mathbf{X}} \in \mathbb{R}^{H \times N}$.

### 2.2 PROBLEM FORMULATION

To address the challenges in MTSF, we first build a mixing model to obtain the main components of each dimension. Secondly we shared these components into sequences by explicit mapping. Finally the equations are solved using optimization equations.

#### 2.2.1 MIXING MODEL

In the temporal mixing model, the observations at any given time step are composed of a mixture of several underlying temporal patterns, referred to as basis functions. According to the model assumption, in multivariate time series data, there exists a set of shared latent temporal basis functions across all channels. Each individual channel is reconstructed by a weighted combination of these basis functions. In order to identify these shared temporal patterns, the model requires channel-wise analysis, which is why the subscript $c$ is used to denote the channel. This process can be interpreted

as channel-wise, as the signal $X[:, n]$ of each channel $n$ is recovered through the weighted combination of the shared temporal basis functions. Specifically, the signal $X_{tn}$ at time step $t$ for channel $n$ can be expressed as:

$$X_{tn} = \sum_{k=1}^{K} A_{c,tk} \cdot S_{c,kn} = (A_{c,t1} \cdot S_{c,1n}) + (A_{c,t2} \cdot S_{c,2n}) + \cdots + (A_{c,tK} \cdot S_{c,Kn}), \quad (1)$$

where the $k$-th term represents the contribution of the $k$-th shared temporal pattern to the final observation. $A_{c,tk} \in T \times k$ denotes the value of the $k$-th temporal basis function (e.g., annual seasonal pattern) at time $t$; and $S_{c,kn} \in k \times N$ represents the static contribution weight of the $k$-th temporal basis function for the $n$-th channel (e.g., oil temperature OT), and is the channel coefficient matrix.. For $T$ time steps, the matrix form of the model can be written as:

$$\mathbf{X} = \mathbf{A}_c \mathbf{S}_c, \quad (2)$$

To ensure physical interpretability, the coefficient matrix must satisfy the following constraints:
• Sum-to-one constraint: The coefficients for each time step must sum to 1, i.e.,

$$\boldsymbol{E}_k^T \mathbf{S}_c = \boldsymbol{E}_T^T, \quad (3)$$

where $\boldsymbol{E}_k = [1, 1, \ldots, 1]^T$ is a column vector with all elements equal to 1, and $[\cdot]^T$ denotes the transpose operation.
• Non-negativity constraint: All elements of the coefficient matrix must be non-negative, i.e.,

$$\mathbf{S}_c \geq 0. \quad (4)$$

Thus, the temporal mixing model can be rewritten as:

$$\mathbf{X} = \mathbf{A}_c \mathbf{S}_c \quad \text{s.t.} \quad \boldsymbol{E}_k^T \mathbf{S}_c = \boldsymbol{E}_T^T, \quad \mathbf{S}_c \geq 0. \quad (5)$$

Similarly, in the channel mixing model, the observed signal is formulated as a dynamic combination of a set of latent channel patterns. This process is represented as:

$$\mathbf{X}^T = \mathbf{A}_t \mathbf{S}_t \quad \text{s.t.} \quad \boldsymbol{E}_k^T \mathbf{S}_t = \boldsymbol{E}_T^T, \quad \mathbf{S}_c \geq 0. \quad (6)$$

Here, the matrix $\mathbf{A}_t \in \mathbb{R}^{N \times k}$ denotes the channel basis functions, where each column represents a fundamental channel structure or group relationship, collectively forming a library of latent factors. The matrix $\mathbf{S}_t \in \mathbb{R}^{k \times T}$ is the temporal coefficient matrix, whose elements define the activation strength of each channel basis function at every time step. The subscript $t$ emphasizes the tick-wise nature of the model, where observations are generated by sequentially activating and integrating distinct channel patterns at each time step. The detailed logic and physical interpretation are provided in Appendix B.1.

### 2.2.2 EXPLICIT MAPPING AND SHARED MECHANISM

The channel-wise and time-wise alignments correspond to two symmetric explicit mapping mechanisms: one maps the channel sequences to shared channel coefficients $S_c$, forming an invariant structure in the time dimension; the other maps the time snapshots to shared channel basis functions $A_t$, forming an invariant structure in the channel dimension. These two paths form the core shared structure of MTS-UNMixers. It is important to note that these two paths reflect a dynamic-static separation within the system. The channel-wise path extracts static attributes that remain stable over time, such as the inherent response pattern of each channel to shared temporal modes. The time-wise path characterizes dynamic evolution, such as the activation level of latent channel factors at each time point. This dynamic-static separation is a key theoretical principle in our model design. Further details and experiments on this dynamic-static separation can be found in Appendix B.1.2.

• In the channel-wise perspective, we treat the complete time series of each channel as a single input token. According to the temporal mixing model (equation 5), multivariate time series are generated by shared temporal basis functions and corresponding channel coefficients. Given any channel $n$, its input token is

$$\mathbf{x}_n = X[:, n] \in \mathbb{R}^T. \quad (7)$$

The encoder executes a Seq2Vec mapping on this sequence. The mapping transforms the input into a vector of size $\mathbb{R}^k$, which corresponds to the static response weights of the channel to the shared temporal basis functions, i.e.,

$$\mathbf{s}_n = f_c(\mathbf{x}_n), \qquad \mathbf{s}_n \in \mathbb{R}^k. \quad (8)$$

All channel coefficient vectors are then collected to form the explicit channel coefficient matrix

$$\mathbf{S}_c = [\,\mathbf{s}_1, \mathbf{s}_2, \ldots, \mathbf{s}_N\,] \in \mathbb{R}^{k \times N}. \tag{9}$$

Since $\mathbf{s}_n$ is derived from the entire historical sequence and does not vary with time, it remains shared across future time steps. Thus, the historical reconstruction and future prediction are given by

$$\mathbf{X}'_{\text{history}} = \mathbf{A}_c \mathbf{S}_c \quad \text{and} \quad \hat{\mathbf{X}}_{\text{future}} = \mathbf{A}_c^{\text{future}} \mathbf{S}_c, \qquad \text{where } \mathbf{A}_c \in \mathbb{R}^{T \times k}, \quad \mathbf{A}_c^{\text{future}} \in \mathbb{R}^{T_{\text{future}} \times k}. \tag{10}$$

In this view, the core of the explicit mapping lies in transforming each channel's time series into a static response vector of size $\mathbb{R}^k$, resulting in the shared channel coefficient matrix $S_c$, which characterizes the static contribution of each channel to the shared temporal modes.

• In the time-wise perspective, we treat each time point's channel snapshot as a single input token. The input sequence comes from the transposed structure of the matrix $X^\top$, i.e.,

$$\mathbf{x}_t = X[t, :] \in \mathbb{R}^N. \tag{11}$$

According to the channel mixing model in equation 6, multivariate time series are generated by shared channel basis functions and corresponding time coefficients. The encoder processes these snapshot sequences along the time dimension, extracting the stable cross-time channel structure patterns from the entire sequence, i.e.,

$$\mathbf{A}_t = f_t(\mathbf{x}_1, \mathbf{x}_2, \ldots, \mathbf{x}_T), \qquad \mathbf{A}_t \in \mathbb{R}^{N \times k}. \tag{12}$$

The corresponding time coefficient matrix is

$$\mathbf{S}_t = [\,\mathbf{s}_1, \mathbf{s}_2, \ldots, \mathbf{s}_T\,] \in \mathbb{R}^{k \times T}. \tag{13}$$

The historical reconstruction and future prediction are given by

$$\mathbf{X}'_{\text{history}} = \mathbf{A}_t \mathbf{S}_t \quad \text{and} \quad \hat{\mathbf{X}}_{\text{future}} = \mathbf{A}_t \mathbf{S}_t^{\text{future}}, \quad \text{where } \mathbf{A}_t \in \mathbb{R}^{N \times k}, \ \mathbf{S}_t^{\text{future}} \in \mathbb{R}^{k \times T_{\text{future}}}. \tag{14}$$

In this view, the core of the explicit mapping lies in extracting the cross-time stable channel structure from the time snapshots, forming the shared channel basis library; **i.e.,** $A_t$ is the channel basis matrix, which characterizes the stable group structures and latent relationships between multiple channels over time.

### 2.2.3 OPTIMIZATION PROCESS

To model feature relationships in historical and future sequences, we estimate the coefficient and basis matrices by minimizing the reconstruction error:

$$(\mathbf{A}, \mathbf{S}) = \arg\min_{\mathbf{A}, \mathbf{S}} \|\mathbf{X} - \mathbf{A}\mathbf{S}\|_1, \tag{15}$$

where $\|\cdot\|_1$ is the L1 norm. By integrating temporal and channel feature extraction, the optimization is formulated as:

$$(\mathbf{A}_c, \mathbf{S}_c, \mathbf{A}_t, \mathbf{S}_t) = \arg\min_{\mathbf{A}_c, \mathbf{S}_c} \|\mathbf{X} - \mathbf{A}_c \mathbf{S}_c\|_1 + \arg\min_{\mathbf{A}_t, \mathbf{S}_t} \|\mathbf{X} - \mathbf{A}_t \mathbf{S}_t\|_1$$
$$\text{s.t.} \quad \boldsymbol{E}_{k_1}^T \mathbf{S}_t = \boldsymbol{E}_T^T, \quad \boldsymbol{E}_{k_2}^T \mathbf{S}_c = \boldsymbol{E}_N^T. \tag{16}$$

With this formulation, we simultaneously extract features along the temporal and channel dimensions and address both reconstruction and prediction within a unified framework. Ultimately, our optimization task is divided into two parts: reconstructing the historical sequence and modeling the future sequence.

## 3 MTS-UNMIXERS NETWORK

### 3.1 OVERALL ARCHITECTURE

Building on Eq. 16, we propose MTS-UNMixers (Fig. 2), which consists of two symmetric paths: the temporal path and the channel path. The temporal path processes each channel time series, using the Mamba network to extract the channel response to shared temporal basis functions, outputting the temporal coefficient matrix $S_c$; the channel path processes each time step channel snapshot, using the Bi-Mamba network to extract the shared channel basis functions $A_t$ and learn the time coefficient matrix $S_t$. These two paths work together through shared basis matrices ($A_t$ and $S_c$), extracting information from the time and channel dimensions, ensuring consistency in the unmixing task and sharing patterns across time and channels, thus achieving a dual unmixing process.

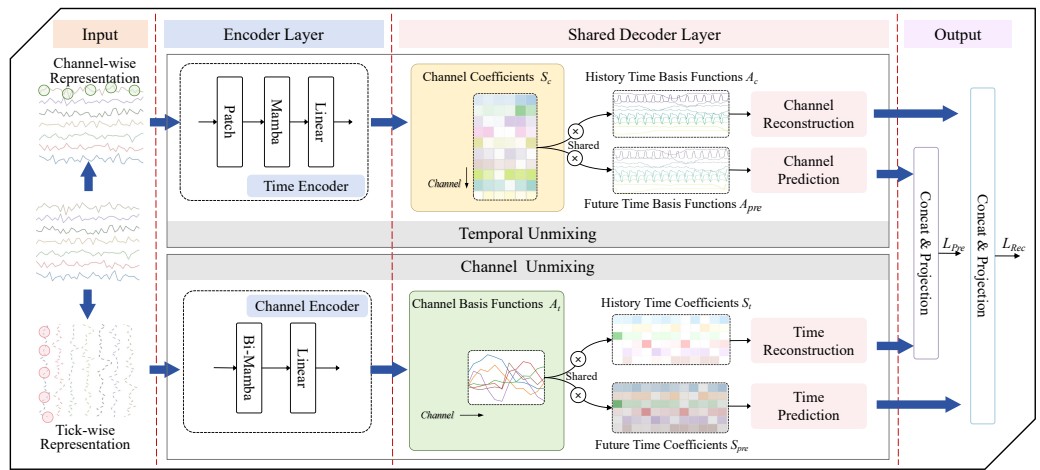

Figure 2: The framework of MTS-UNMixers. The input series is projected into channel-wise and tick-wise representations. The time branch employs Mamba for temporal encoding, while the channel branch uses Bi-Mamba to model inter-variable relations. Both branches perform reconstruction and prediction with shared modules, and their outputs are concatenated for joint optimization of $L_{Rec}$ and $L_{Pre}$.

## 3.2 UNMIXING ENCODER

**Temporal (Channel-wise Representation) Unmixing Encoder.** The Time Encoder processes the historical sequence by dividing it into segments $\mathcal{X}_p \in \mathbb{R}^{N \times T_p \times P}$. These tokens are processed through a Mamba block (see Appendix for details) to capture long-range temporal dependencies and nonlinear dynamics efficiently. The output is computed as:

$$\mathcal{X}_m = \mathbf{SSM}\left(\mathbf{Conv}(\mathbf{Linear}(\mathcal{X}_p))\right) * \sigma(\mathbf{Linear}(\mathcal{X}_p)), \tag{17}$$

where $\sigma$ is a nonlinear activation function. Finally, a linear layer with softmax generates the weight matrix:

$$\mathbf{S}_c = \mathbf{Softmax}(\mathbf{Linear}(\mathcal{X}_m)). \tag{18}$$

**Channel (Tick-wise Representation) Unmixing Encoder.** The channel encoder employs a bidirectional Mamba block (see Appendix) to process inter-channel dependencies at each timestep (Fig. 3b). Unlike causal temporal modeling, channel interactions require bidirectional context awareness, which the Mamba block achieves through nonlinear feature extraction. Specifically, the forward Mamba processes the sequence $\mathbf{X}$ in its original order, while the backward Mamba processes the reversed sequence. Key operations include:

$$\mathbf{X}_e = \mathbf{Linear}\left(\mathbf{Mamba}_{fw}(\mathbf{X}) + \mathbf{Mamba}_{bw}(\mathbf{Reverse}(\mathbf{X}))\right), \tag{19}$$

where $\mathbf{X}$ denotes the linearly transformed historical sequence. The output $\mathbf{X}_e$ generates the channel basis function $\mathbf{A}_t$, encoding primary temporal patterns with ReLU activation for enhanced nonlinearity.

$$\mathbf{X}' = \mathbf{Linear}(\mathbf{Concat}(\mathbf{X}'_c, \mathbf{X}'_t)). \tag{20}$$

The linear layer processes the fused data to generate the final reconstructed sequence.

## 3.3 SHARED DECODER LAYER

The decoder reconstructs historical sequences and predicts future sequences through dual-path feature projection, maintaining consistency via shared components. Based on the content of the previous section, the following is obtained:

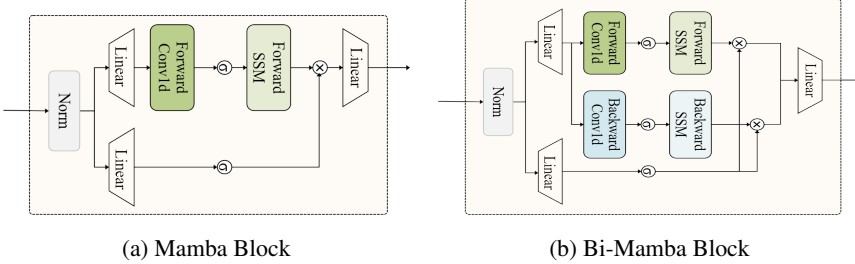

(a) Mamba Block          (b) Bi-Mamba Block

Figure 3: Comparison of (a) Mamba Block and (b) Bi-Mamba Block structures.

**Channel Reconstruction and Prediction.** Using the softmax-normalized coefficient matrix $\mathbf{S}_c$ from the Time Encoder, we compute:

$$\mathbf{X}'_c = \mathbf{A}_c\,\mathbf{S}_c, \quad \hat{\mathbf{X}}_c = \mathbf{A}_p\,\mathbf{S}_c, \tag{21}$$

where $\mathbf{A}_c$ and $\mathbf{A}_p$ are learnable reconstruction/prediction basis matrices. This shared $\mathbf{S}_c$ design preserves cross-phase feature alignment while enabling task-specific representation learning.

**Temporal Reconstruction and Prediction.** The Channel Encoder generates shared basis matrix $\mathbf{A}_t$, representing core temporal patterns. We compute:

$$\mathbf{X}'_t = \mathbf{A}_t\,\mathbf{S}_t, \quad \hat{\mathbf{X}}_t = \mathbf{A}_t\,\mathbf{S}_p, \tag{22}$$

where $\mathbf{S}_t$ and $\mathbf{S}_p$ are softmax-constrained coefficient matrices for reconstruction/prediction respectively. The shared $\mathbf{A}_t$ ensures temporal pattern continuity between historical and future sequences through consistent feature embedding.

**Projection Layer.** In obtaining the prediction results from the temporal and channel axes, the two are concatenated, and a linear layer is applied to generate the final output:

$$\hat{\mathbf{X}} = \mathbf{Linear}(\mathbf{Concat}(\hat{\mathbf{X}}_c, \hat{\mathbf{X}}_t)). \tag{23}$$

Through concatenation and linear transformation, the model integrates features from both dimensions to enhance prediction accuracy. Similarly, the reconstructed data undergoes fusion to obtain the reconstructed historical sequence:

### 3.4 LOSS FUNCTION

Based on the optimization formulas 16 presented earlier, we employ the L1 loss function to solve the sequence reconstruction and prediction tasks by measuring the absolute differences between predicted and actual values. Compared to L2 loss, L1 loss is more robust to outliers, making it better suited for capturing the main patterns and trends in time series data. The overall L1 loss function, incorporating both reconstruction and prediction objectives, is defined as follows:

$$\mathcal{L} = \lambda_1 \cdot \|\mathbf{X}' - \mathbf{X}_{\text{history}}\|_1 + \lambda_2 \cdot \|\hat{\mathbf{X}} - \mathbf{X}_{\text{future}}\|_1, \tag{24}$$

where $\lambda_1$ and $\lambda_2$ are weighting factors that balance the importance of the reconstruction and prediction tasks.

## 4 EXPERIMENT RESLUTS

### 4.1 DATASETS DESCRIPTIONS

We conducted experiments on seven real-world datasets: ETT (ETTh1, ETTh2, ETTm1, ETTm2), Weather, Traffic, and Electricity. We followed the same data processing protocol and prediction horizon settings as used in TimesNet Wu et al. (2023), varying the forecasting length among $\{96, 192, 336, 720\}$ to assess model performance across different forecast windows.

Table 1: Multivariate time series forecasting results (averaged over all horizons). The best performance is highlighted in **red**, and the second-best is blue. Full results are in Appendix G.

| Model | ETTh1 | | ETTh2 | | ETTm1 | | ETTm2 | | Weather | | Traffic | | Electricity | | 1st Count |
|---|---|---|---|---|---|---|---|---|---|---|---|---|---|---|---|
| | MSE | MAE | MSE | MAE | MSE | MAE | MSE | MAE | MSE | MAE | MSE | MAE | MSE | MAE | |
| Ours | **0.423** | **0.417** | **0.345** | **0.379** | **0.375** | **0.382** | **0.270** | **0.314** | **0.237** | **0.261** | 0.466 | _0.292_ | **0.176** | **0.269** | 12 |
| TimeXer | _0.431_ | _0.432_ | 0.378 | 0.403 | _0.375_ | _0.391_ | 0.278 | 0.323 | 0.254 | 0.276 | 0.447 | 0.295 | 0.180 | 0.274 | 0 |
| TimeMixer | 0.447 | 0.440 | _0.365_ | _0.395_ | 0.381 | 0.396 | _0.275_ | _0.322_ | _0.240_ | _0.272_ | 0.485 | 0.298 | 0.182 | 0.273 | 0 |
| TimeMachine | 0.439 | 0.439 | 0.387 | 0.419 | 0.399 | 0.407 | 0.276 | 0.335 | 0.252 | 0.284 | **0.425** | 0.307 | 0.183 | 0.272 | 1 |
| iTransformer | 0.454 | 0.448 | 0.383 | 0.407 | 0.407 | 0.410 | 0.288 | 0.332 | 0.258 | 0.278 | _0.428_ | **0.282** | _0.178_ | _0.270_ | 1 |
| SiMBA | 0.473 | 0.455 | 0.452 | 0.448 | 0.444 | 0.466 | 0.338 | 0.370 | 0.255 | 0.272 | 0.513 | 0.422 | 0.199 | 0.271 | 0 |
| PatchTST | 0.451 | 0.441 | 0.366 | 0.395 | 0.384 | 0.396 | 0.285 | 0.328 | 0.258 | 0.281 | 0.511 | 0.317 | 0.203 | 0.284 | 0 |
| TimesNet | 0.458 | 0.450 | 0.414 | 0.427 | 0.400 | 0.406 | 0.291 | 0.333 | 0.259 | 0.287 | 0.620 | 0.336 | 0.193 | 0.295 | 0 |
| FITS | 0.714 | 0.576 | 0.422 | 0.433 | 0.718 | 0.563 | 0.321 | 0.358 | 0.293 | 0.322 | 1.439 | 0.810 | 0.862 | 0.766 | 0 |
| DLinear | 0.456 | 0.465 | 0.559 | 0.547 | 0.403 | 0.407 | 0.350 | 0.401 | 0.265 | 0.317 | 0.625 | 0.383 | 0.212 | 0.300 | 0 |
| FEDformer | 0.440 | 0.460 | 0.434 | 0.447 | 0.448 | 0.452 | 0.305 | 0.349 | 0.309 | 0.360 | 0.573 | 0.347 | 0.205 | 0.315 | 0 |
| TiDE | 0.518 | 0.517 | 0.387 | 0.419 | 0.413 | 0.415 | 0.293 | 0.338 | 0.271 | 0.320 | 0.608 | 0.377 | 0.209 | 0.295 | 0 |
| Stationary | 0.570 | 0.537 | 0.526 | 0.516 | 0.481 | 0.456 | 0.306 | 0.347 | 0.288 | 0.314 | 0.624 | 0.340 | 0.193 | 0.296 | 0 |
| Autoformer | 0.496 | 0.487 | 0.450 | 0.459 | 0.588 | 0.517 | 0.327 | 0.371 | 0.338 | 0.314 | 0.628 | 0.340 | 0.227 | 0.338 | 0 |

## 4.2 MAIN RESULTS

We compared MTS-UNMixers with representative models across different architectures, including Transformer-based methods (PatchTSTNie et al. (2023), FEDformerZhou et al. (2022), AutoformerWu et al. (2021), iTransformerLiu et al. (2023), Stationary TransformerLiu et al. (2022b), TimeXerWang et al. (2024b)), MLP-based methods (DLinearZeng et al. (2023), FITSXu et al. (2023), TiDEDas et al. (2023)), and CNN-based methods (TimesNetWu et al. (2023), TimeMixerWang et al. (2024a)), and Mamba-based methods (TimemachineAhamed & Cheng (2024a), SiMBAPatro & Agneeswaran (2024b)). As shown in Table 1, MTS-UNMixers achieves top-2 average rankings across all seven datasets, with detailed results reported in the Appendix.

## 4.3 ABLATION STUDY

In the ablation study, we systematically evaluated the importance of the main modules within MTS-UNMixers in time-series forecasting, as shown in Table 2.

The results show that removing these modules caused significant performance drops, particularly for long-term predictions. For example, on the ETTh1 dataset, the MSE for the 96-step forecast increased by about 1.9% after removing the Channel Unmixing module. The MSE dropped by 29.6% when the Time Unmixing module was removed, emphasizing its critical role in reducing aliasing errors and capturing temporal dependencies. Removing Mamba led to a notable drop in performance for long-term forecasts, underlining its importance in capturing nonlinear causal relationships and long-term trends. Removing Bi-Mamba caused performance degradation, especially on the Weather dataset with complex inter-channel interactions, showing its role in modeling variable relationships.

Table 2: Performance comparison of MTS-UNMixers and its ablated versions on ETTh1, ETTm1, and Weather datasets at various prediction lengths.

| Dataset | Length | MTS-UNMixers | | w/o Channel Unmixing | | w/o Time Unmixing | | w/o Mamba | | w/o Bi-Mamba | |
|---|---|---|---|---|---|---|---|---|---|---|---|
| | | MSE | MAE | MSE | MAE | MSE | MAE | MSE | MAE | MSE | MAE |
| ETTh1 | 96 | **0.368** | **0.388** | 0.375 | 0.393 | 0.477 | 0.458 | 0.369 | 0.393 | 0.374 | 0.391 |
| | 192 | **0.427** | **0.419** | 0.433 | 0.425 | 0.518 | 0.479 | 0.439 | 0.430 | 0.433 | 0.428 |
| | 336 | **0.443** | **0.433** | 0.454 | 0.442 | 0.537 | 0.498 | 0.454 | 0.443 | 0.449 | 0.438 |
| | 720 | **0.454** | **0.429** | 0.466 | 0.436 | 0.547 | 0.516 | 0.461 | 0.448 | 0.460 | 0.437 |
| ETTm1 | 96 | **0.317** | **0.350** | 0.321 | 0.355 | 0.383 | 0.364 | 0.320 | 0.351 | 0.319 | 0.353 |
| | 192 | **0.360** | **0.378** | 0.366 | 0.384 | 0.478 | 0.418 | 0.375 | 0.389 | 0.366 | 0.381 |
| | 336 | **0.388** | **0.400** | 0.396 | 0.406 | 0.476 | 0.441 | 0.411 | 0.418 | 0.396 | 0.407 |
| | 720 | **0.442** | **0.422** | 0.461 | 0.432 | 0.560 | 0.530 | 0.465 | 0.449 | 0.460 | 0.435 |
| Weather | 96 | **0.152** | **0.189** | 0.159 | 0.196 | 0.204 | 0.270 | 0.162 | 0.226 | 0.163 | 0.200 |
| | 192 | **0.201** | **0.237** | 0.220 | 0.252 | 0.261 | 0.284 | 0.218 | 0.260 | 0.213 | 0.248 |
| | 336 | **0.254** | **0.286** | 0.264 | 0.295 | 0.379 | 0.372 | 0.275 | 0.294 | 0.260 | 0.287 |
| | 720 | **0.339** | **0.336** | 0.346 | 0.350 | 0.471 | 0.372 | 0.352 | 0.356 | 0.342 | 0.345 |

Mamba has a greater impact on long-sequence forecasting, while Bi-Mamba excels at handling multivariable interactions. Together, they complement each other to enhance accuracy and robustness. Overall, the full MTS-UNMixers model consistently outperformed others, demonstrating the importance of each module in improving prediction accuracy.

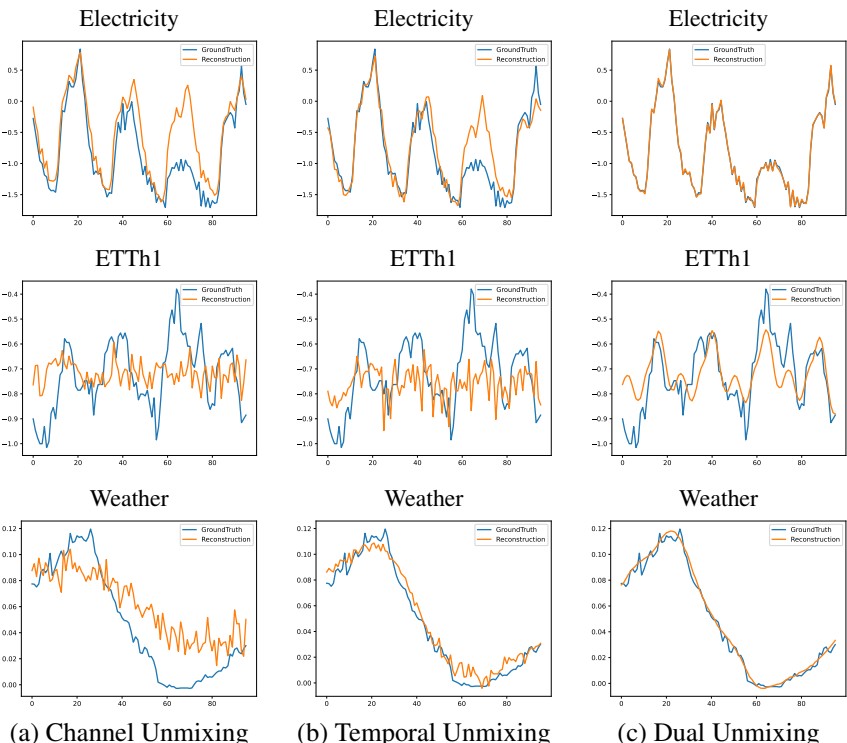

(a) Channel Unmixing    (b) Temporal Unmixing    (c) Dual Unmixing

Figure 4: Reconstruction visualization of MTS-UNMixers under different configurations on ETTh1, Weather, and Electricity datasets using a history sequence length of 96. (a) Channel Unmixing Only, (b) Temporal Unmixing Only, (c) Dual Unmixing. Dual-path unmixing achieves the best reconstruction accuracy by effectively capturing both temporal dependencies and inter-channel relationships.

## 4.4 RECONSTRUCTION VISUALIZATION

To better understand the performance of MTS-UNMixers, we designed reconstruction experiments with different module configurations and conducted prediction visualization on the electricity dataset. These experiments utilized a history sequence length of 96 for the Weather, and Electricity datasets, with the following configurations:

**Channel Unmixing Only:** In this setup, we retained only the channel unmixing module to reconstruct the input data. As shown in Fig. 4 (a), channel unmixing effectively captures inter-channel relationships, allowing feature extraction along the channel dimension. However, without temporal unmixing, the model struggles with time dependencies, leading to visible discrepancies in trend continuity, especially in datasets with long-term dependencies.

**Temporal Unmixing Only:** In this setup, only the temporal unmixing module was used. Fig. 4 (b) shows that temporal unmixing better captures time dependencies and dynamic trends, resulting in more accurate temporal reconstructions. However, it lacks inter-channel feature extraction, leading to weaker reconstruction of inter-variable interactions.

**Dual Unmixing:** In this complete configuration, both temporal and channel unmixing modules were applied. As illustrated in Fig. 4 (c), dual unmixing enables the model to deeply extract features across both dimensions, capturing both temporal dependencies and inter-channel details with

high accuracy. This dual-path setup demonstrates superior reconstruction compared to single-path configurations.

Comparative analysis shows that dual-path unmixing achieves superior information extraction and reconstruction. While channel-only unmixing captures channel-specific features but misses temporal dependencies, and temporal-only unmixing neglects inter-channel details, combining both yields optimal accuracy and integrity, validating the design of MTS-UNMixers.

## 5 CONCLUSION

In conclusion, this paper presents MTS-UNMixers, a novel approach to time-series forecasting that leverages unmixing and sharing mechanisms within a Mamba-based network. By using Mamba blocks to separate channel coefficients and temporal basis functions, MTS-UNMixers captures complex inter-channel relationships and temporal dependencies with high precision. The integration of these unmixing mechanisms with a sharing phase enables efficient mapping of historical patterns to future predictions, effectively addressing challenges related to signal aliasing and redundancy. Extensive experiments on seven public datasets demonstrate that the model achieves superior performance compared to nine state-of-the-art baselines in various long-term forecasting tasks. The robust performance of MTS-UNMixers highlights its ability to effectively model intricate temporal dynamics and dependencies, offering significant advancements for multivariate time-series forecasting applications.

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

# A    RELATED WORK

## A.1    DEEP LEARNING MODELS FOR TIME SERIES FORECASTING

In recent years, deep learning has been widely applied in time series forecasting, with models broadly categorized into attention-based and non-attention-based approaches. Attention-based models primarily include the Transformer series, which utilize self-attention mechanisms to capture long-range dependencies and relationships between different time points in time series data. These models are particularly suited for forecasting tasks involving long sequences and high-dimensional data Nie et al. (2023); Zhou et al. (2022); Wu et al. (2021); Liu et al. (2022b). Nie et al. proposed PatchTST Nie et al. (2023), a Transformer-based model that enhances efficiency and long-term forecasting accuracy by leveraging time series segmentation and channel independence. Zhou et al. introduced FEDformer Zhou et al. (2022), which combines Transformer with seasonal-trend decomposition and frequency enhancement to capture both global structures and detailed features, improving long-term forecasting performance.

Non-attention-based models include recurrent neural network (RNN)-based, convolutional neural network (CNN)-based, Mamba-based, and multi-layer perceptron (MLP)-based models. These models capture local dependencies, time series characteristics, and multi-scale features through various approaches. MLP-based models apply multi-layer perceptrons along the temporal dimension, offering simplicity and competitive performance. For example, Wang et al. proposed TimeMixer Wang et al. (2024a), a fully MLP-based architecture that disentangles and captures multi-scale temporal variations to enhance forecasting performance. CNN-based models leverage convolutional kernels to capture local patterns over time. Liu et al. proposed SCINet Liu et al. (2022a), which recursively downsamples and convolves to model complex temporal dynamics effectively. RNN-based models manage temporal dependencies through recurrent structures. For instance, DeepAR Salinas et al. (2020) achieves high-precision probabilistic forecasting by training an auto-regressive recurrent network on a large collection of related time series.

## A.2    STATE SPACE MODELS

State Space Models (SSMs) Rangapuram et al. (2018); Lin et al. (2021); Gu et al. (2021); Gu & Dao (2023) utilize intermediate state variables for sequence-to-sequence mapping, enabling efficient handling of long sequences. By capturing complex dynamic features through state variables, SSMs address computational and memory bottlenecks in long-sequence modeling. However, earlier SSMs often required high-dimensional matrix operations, resulting in significant computational demands. Rangapuram et al. proposed Deep State Space Models (DSSM) Rangapuram et al. (2018), integrating latent states with deep neural networks for better dynamic modeling. Lin et al. introduced SSDNet Lin et al. (2021), combining Transformers with SSMs for temporal pattern learning while avoiding Kalman filters. Structured state-space models like S4 Gu et al. (2021) improved upon these limitations by employing low-rank corrections to stabilize diagonalization and reduce SSMs to a simplified Cauchy kernel. This approach significantly lowered computational and memory requirements while maintaining performance, making SSMs more practical for long-sequence modeling.

Building on these advances, Gu et al. proposed Mamba Gu & Dao (2023), which integrates parameterized matrices with hardware-aware parallel computing to achieve superior efficiency. Several derivatives of Mamba further improved time series forecasting. For instance, Ma et al. introduced FMamba Ma et al. (2024), which combines fast attention with Mamba for efficient dependency modeling. Patro and Agneeswaran proposed SiMBA Patro & Agneeswaran (2024a), which simplifies Mamba by incorporating EinFFT for channel modeling, achieving strong performance in image and time series benchmarks. Tang et al. Tang et al. (2024) introduced VMRNN, which incorporates Vision Mamba blocks with LSTM for spatiotemporal forecasting. Ahamed and Cheng developed TimeMachine Ahamed & Cheng (2024b), a quadruple-Mamba architecture that excels in accuracy, scalability, and efficiency. While Mamba has achieved notable progress, its full potential in complex prediction tasks remains unexplored, warranting further refinement.

## B  INTERPRETABILITY ANALYSIS AND VISUALIZATION VALIDATION

### B.1  PHYSICAL FOUNDATIONS OF THE MIXING ASSUMPTION

#### B.1.1  MIXING MODEL

In many real-world systems, multivariate time series do not evolve independently; instead, they are jointly influenced by a smaller number of more fundamental latent drivers. Examples include:

- Power systems: oil temperature, transformer load, residential demand, and industrial demand are jointly affected by seasonal temperature and weekday–weekend rhythms.
- Transportation systems: traffic flows on different roads are simultaneously driven by commuting cycles, holiday effects, and unexpected events.
- Economic systems: stock returns, sector indices, and trading volumes are jointly influenced by macro factors such as risk appetite, policy expectations, and interest-rate conditions.
- Climate time series: temperature, humidity, and wind speed are jointly driven by seasonal cycles, diurnal cycles, and regional geographic structures.

In such systems, the pattern of shared drivers and channel-specific responses is ubiquitous. Therefore, the mixing formulations

$$\mathbf{X} = \mathbf{A}_c\,\mathbf{S}_c, \quad \mathbf{X}^\top = \mathbf{A}_t\,\mathbf{S}_t \tag{25}$$

are not merely mathematical abstractions but structural descriptions of how real-world multivariate systems behave.

#### B.1.2  SEPARATION OF DYNAMICS AND STATIC COMPONENTS

Building on this, multivariate systems typically exhibit a separation of dynamics and static components structure, where the system contains both structurally stable attributes that remain constant over time and behavioral patterns that evolve with time.

**1. Static invariants:** Physical properties that remain stable over time (shared across channels).

**- Channel basis structures** $\mathbf{A}_t \in \mathbb{R}^{N \times k}$, representing channel factors or groups that remain stable over time. These include:

- In power systems: Temperature-related variables (OT, load) form a "temperature factor," industrial and commercial loads form a "business factor," and residential load forms a "lifestyle factor."
- In economic systems: Major market indices, cyclical stocks, and growth stocks form a "macro market factor," while bank and insurance stocks form an "interest rate-sensitive factor."
- In transportation systems: Highway traffic, national road traffic, and city main road traffic form a "commuting factor," while tourist routes form a "leisure factor."

These structures represent long-term arrangements and combinations of variables that do not change due to short-term fluctuations, and they participate in the sharing of common drivers.

**- Channel Contribution Coefficients** $\mathbf{S}_c \in \mathbb{R}^{k \times N}$, representing the inherent sensitivity and response characteristics of each channel to the temporal basis patterns. These are used to describe each channel's inherent response to the temporal basis patterns. Examples include:

- Temperature-type response: Oil temperature (OT) and high load (HUFL) will respond significantly during high summer temperatures.
- Lagged response: For example, HULL is influenced by the lag effect of HUFL, leading to a "phase inversion" in the annual cycle.
- Inverse coupling relationship: The response direction of humidity (HU) and temperature (OT) is often opposite.

- Industry characteristics: Manufacturing electricity usage fluctuates significantly with workdays, while residential electricity usage is smoother.

These are inherent channel properties that do not change over time and participate in the sharing process.

**2. Dynamic Variables:** System states that change over time (non-shared).

**- Temporal Basis Patterns** $\mathbf{A}_c \in \mathbb{R}^{T \times k}$, describing the various time-driven dynamics exhibited by the system within the observation window (e.g., seasonality, weekly cycles, trends). Specific time-driven dynamics that the system undergoes during the observation period, for example:

- Annual seasonality: High summer temperatures, winter heating.
- Weekly cycles: Workday and weekend patterns.
- Trend changes: Long-term rise or decline in certain industries.
- **Short-term fluctuations:** Cold fronts, heatwaves, holiday effects.

These patterns change with the time window and therefore cannot be shared.

**- Temporal Activity** $\mathbf{S}_t \in \mathbb{R}^{k \times T}$, representing the activation strength of each channel factor at each time point. Examples include:

- The "temperature factor" has the highest activation in summer each year.
- The "business electricity factor" activates during weekdays and decreases on weekends.
- The "macro-economic factor" experiences instant activation on policy announcement days.
- The "tourist traffic factor" is strongly activated during holidays.

These describe the dynamic evolution over time and are therefore non-shared.

## B.2 THE NECESSITY OF MIXING MODEL AND SHARED STRUCTURE

Under the mixing assumption and the separation of dynamics and static components framework, the design of the two unmixing paths is uniquely determined by the semantics of the input tokens: the complete time series of a channel only contains the inherent response information of the channel, so its encoding result must be the time-stable channel contribution matrix $S_c$; a system snapshot at a single point in time contains the long-term invariant structural relationships between channels, so its encoding result must be the time-shared channel basis matrix $A_t$. These two static components together form the structural knowledge base of the system and are shared within the model. Table 3 summarizes the model logic of the two unmixing paths.

| Perspective (-wise) | Channel-wise (Temporal Unmixing) | Time-wise (Channel Unmixing) |
|---|---|---|
| **Input Token** | Single channel complete time series $X[:, j]$ | Single time-step full channel snapshot $X[t, :]$ |
| **Encoder Objective** | Extract inherent response to temporal bases | Extract time-invariant structure and factor grouping |
| **Encoder Structure** | Mamba | Bi-Mamba |
| **Output (Shared)** | $S_c \in \mathbb{R}^{k \times N}$ | $A_t \in \mathbb{R}^{N \times k}$ |
| **Physical Meaning** | Inherent physical individuality, sensitivity, and lag of the channel | Long-term stable structural relationships and factor patterns across channels |

Table 3: Unmixing Paths and Their Physical Interpretations

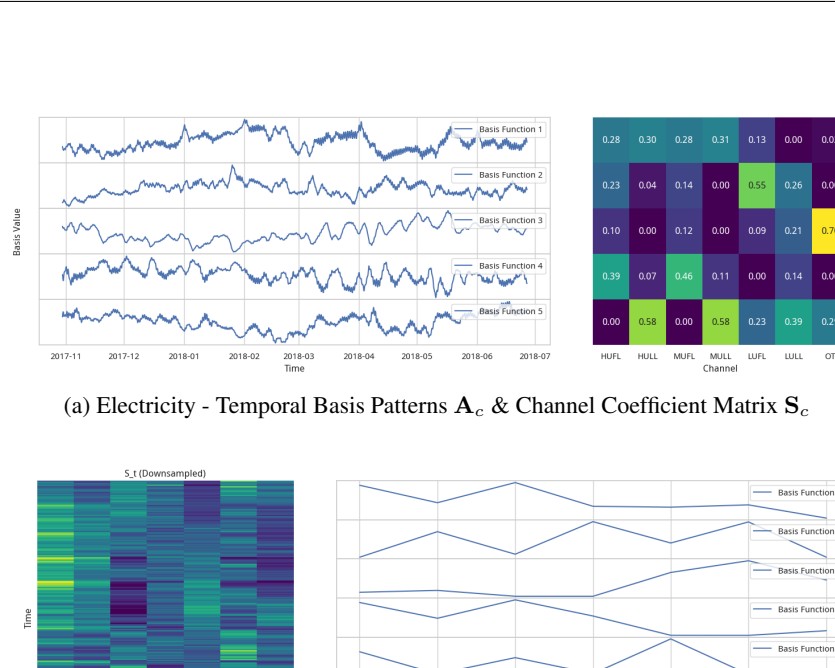

(a) Electricity - Temporal Basis Patterns $\mathbf{A}_c$ & Channel Coefficient Matrix $\mathbf{S}_c$

(b) Electricity - Temporal Coefficient Matrix $\mathbf{S}_t$ & Channel Basis Patterns $\mathbf{A}_t$

Figure 5: The Electricity dataset reveals the intrinsic rhythms and structures of physical systems.

## B.3 EXPERIMENTAL VISUALIZATION

To validate the physical interpretability of the unmixing results from MTS-UNMixers, we conduct a visualization analysis on the ETTh1 and Electricity datasets. We analyze the Time Unmixing and Channel Unmixing paths separately and, based on the physical meaning of the shared matrices, explain the time patterns, channel structures, and their ability to reveal the real system behavior extracted by the model.

### B.3.1 ETTH1 ANALYSIS

- **Temporal unmixing (Figure 5a).** The five curves in the shared temporal bases ($A_c$) represent the fundamental temporal rhythms that all channels follow. These basis functions exhibit diverse shapes: some curves, such as Basis Function 1, show smooth long-term trends, while others display clear periodic patterns or irregular fluctuations. This demonstrates that the model is capable of separating multiple types of temporal dynamics embedded in the raw data, including long-term trends, seasonality, and short-term variations. The heatmap of the channel contribution coefficients ($S_c$) quantifies the extent to which each physical channel (such as HUFL and OT) participates in each temporal rhythm. For example, if the block corresponding to OT is the brightest in the row associated with Basis Function 3, it indicates that the variation in transformer oil temperature (OT) primarily follows the third temporal rhythm. This visualization confirms that the model can learn a stable contribution matrix that captures the inherent physical characteristics of each channel.

- **Channel unmixing (5b).** Each curve in the shared channel bases ($A_t$) represents a latent physical process or functional group identified by the model. For instance, if one curve (such as Basis Function 2) has high values for HUFL and HULL but low values for other variables, it indicates that the model has successfully identified a distinct high-utilization load process. More importantly, if a curve exhibits consistently high values across all variables, it corresponds to a common factor that drives all channels, such as ambient temperature, which is the central value of unmixing analysis. The corresponding heatmap

of temporal activity ($S_t$) shows the activation intensity of each physical process at every time point. A continuous horizontal bright band indicates that a process remains active throughout the entire period, while intermittent vertical bright bands indicate that a process is activated only during specific time intervals.

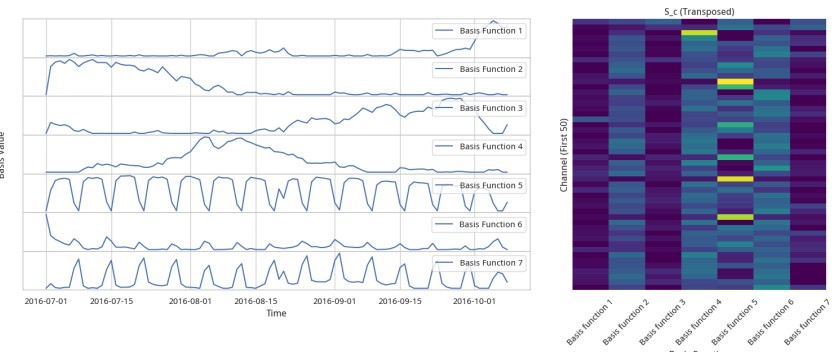

(a) ETTh1 - Temporal Basis Patterns $\mathbf{A}_c$ & Channel Coefficient Matrix $\mathbf{S}_c$

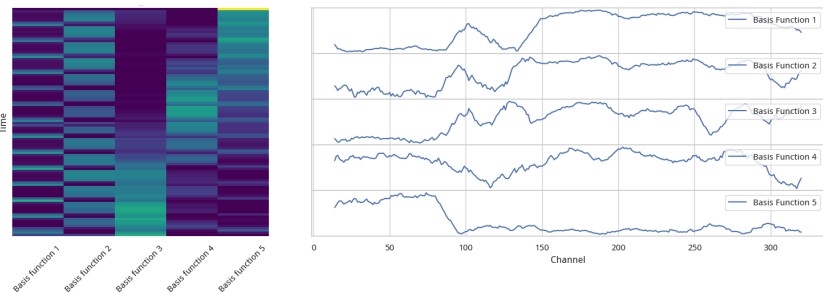

(b) ETTh1 - Temporal Coefficient Matrix $\mathbf{S}_t$ & Channel Basis Patterns $\mathbf{A}_t$

Figure 6: The ETTh1 dataset reveals the intrinsic rhythms and structures of physical systems.

### B.3.2 ELECTRICITY ANALYSIS

- **Temporal unmixing (Figure 6a).** The curves in the shared temporal bases ($A_c$) represent the temporal variation patterns experienced collectively by all electricity users. Since this is electricity consumption data, it is expected that the model captures patterns closely related to human activity. With a sufficiently long time span, clear daily, weekly, and annual periodicities should emerge. The regular fluctuations observable in the figure, even within a window of only one hundred days, confirm that the model is learning these shared consumption rhythms. The heatmap of the channel contribution coefficients ($S_c$), shown in transposed form, illustrates the response strength of the first fifty users to each temporal pattern. User clustering can be observed from this visualization: for example, if the pattern of a given row (representing a specific user) is highly similar to several other rows, it indicates that they belong to the same type of consumption group, such as residential users or industrial users. This verifies that the model is able to learn personalized response coefficients for each channel.

- **Channel unmixing (6b).** Each curve in the shared channel bases ($A_t$) defines a distinct user group. For example, Basis Function 1 may show high values for channels corresponding to residential users and low values for channels corresponding to industrial users, thereby achieving data-driven user classification. The heatmap of temporal activity ($S_t$) shows how the overall consumption behavior of these user groups evolves over time. For instance, the row corresponding to the industrial user group may exhibit a clear periodic decline in activation during weekends.

## C TECHNICAL DETAILS OF MAMBA BLOCKS

### C.1 MAMBA ARCHITECTURE

The Mamba block implements a parameterized state space model (SSM) with hardware-aware optimizations. Its mathematical formulation consists of three phases:

#### C.1.1 CONTINUOUS STATE SPACE MODEL

The continuous-time SSM is defined by:

$$
\begin{aligned}
h'(t) &= \tilde{A}(x(t))h(t) + \tilde{B}(x(t))x(t), \\
y(t) &= \tilde{C}(x(t))h(t),
\end{aligned}
\tag{26}
$$

where:

- $\tilde{A} \in \mathbb{C}^{N \times N}$: State transition matrix *dynamically generated* from input $x(t)$
- $\tilde{B} \in \mathbb{C}^{N \times d}$: Input projection matrix conditioned on $x(t)$
- $\tilde{C} \in \mathbb{C}^{d \times N}$: Output projection matrix adapted to $x(t)$
- $N$: State dimension, $d$: Input dimension

#### C.1.2 DISCRETIZATION

Using zero-order hold (ZOH) with input-dependent step size $\Delta(x)$, we discretize the system:

$$
\begin{aligned}
\bar{A} &= e^{\Delta \tilde{A}}, \\
\bar{B} &= (\Delta \tilde{A})^{-1}(e^{\Delta \tilde{A}} - I)\Delta \tilde{B}, \\
h_k &= \bar{A}h_{k-1} + \bar{B}x_k, \\
y_k &= \bar{C}h_k,
\end{aligned}
\tag{27}
$$

where $\Delta = \text{Softplus}(\text{Linear}(x))$ ensures positive step sizes.

#### C.1.3 PARAMETER GENERATION

The SSM parameters are dynamically produced via:

$$
\begin{aligned}
(\tilde{A}, \tilde{B}, \tilde{C}) &= \text{ParamLayers}(X_{\text{proj}}), \\
X_{\text{proj}} &= \text{Linear}(X) \in \mathbb{R}^{L \times n},
\end{aligned}
\tag{28}
$$

where ParamLayers contains:

- $\tilde{A}$: Diagonal matrix initialized via HiPPO Gu et al. (2020)
- $\tilde{B}, \tilde{C}$: Dense projections with learned biases

#### C.1.4 HARDWARE OPTIMIZATION

To enable efficient computation:

$$
\tilde{A} = \underbrace{\text{Diag}(\lambda_1, ..., \lambda_N)}_{\text{Diagonal}} + \underbrace{UV^T}_{\text{Low-rank}}, \quad U, V \in \mathbb{C}^{N \times r},
\tag{29}
$$

where $r \ll N$ (typically $r = 1$). This allows:

- $O(N)$ memory complexity via diagonal dominance
- Parallel scan algorithms for $O(L)$ time complexity Dao et al. (2024)

## C.2 BIDIRECTIONAL MAMBA (BIMAMBA)

### C.2.1 ARCHITECTURE DESIGN

BiMamba processes sequences bidirectionally:

$$
\begin{aligned}
Y_{\text{fw}} &= \text{Mamba}(X; \{\tilde{A}_{\text{fw}}, \tilde{B}_{\text{fw}}, \tilde{C}_{\text{fw}}\}), \\
Y_{\text{bw}} &= \text{Mamba}(\text{Reverse}(X); \{\tilde{A}_{\text{bw}}, \tilde{B}_{\text{bw}}, \tilde{C}_{\text{bw}}\}),
\end{aligned}
\tag{30}
$$

where:

- Reverse($\cdot$): Temporal reversal operation along sequence dimension

- Parameter independence: $\{\tilde{A}, \tilde{B}, \tilde{C}\}_{\text{fw/bw}}$ generated separately

### C.2.2 FEATURE FUSION

The bidirectional features are combined via:

$$
Y_{\text{bi}} = \text{Linear}(Y_{\text{fw}} \oplus Y_{\text{bw}}),
\tag{31}
$$

where $\oplus$ denotes either:

- Summation: $Y_{\text{fw}} + Y_{\text{bw}}$ (default)

- Concatenation: $[Y_{\text{fw}}; Y_{\text{bw}}]$ (requires 2x output dimension)

### C.2.3 GRADIENT STABILITY

To prevent gradient explosion/vanishing:

$$
\frac{\partial \mathcal{L}}{\partial Y_{\text{bi}}} = \text{Linear}^T \left( \frac{\partial \mathcal{L}}{\partial Y_{\text{bi}}} \right) \odot \left( \frac{\partial (Y_{\text{fw}} \oplus Y_{\text{bw}})}{\partial Y_{\text{fw/bw}}} \right),
\tag{32}
$$

where the linear layer acts as gradient stabilizer.

## D EXPERIMENTAL ENVIRONMENT

Our experiments were conducted on a high-performance workstation equipped with an Intel Core i9-13900K CPU (24 cores, 32 threads), an NVIDIA GeForce RTX 4090 GPU (24 GB VRAM), 128 GB of DDR5 RAM, and a 2 TB NVMe SSD. Our software environment included Ubuntu 22.04 LTS as the operating system, CUDA 11.8, cuDNN 8.6.0, Python 3.10.12, and PyTorch 2.0.1. We used the latest versions of the `causal-conv1d` and `mamba` libraries to implement and evaluate our models.

## E DATASET DETAILS

### E.1 ETT DATASET

- **Source**: Data from two power transformers at distinct sites.

- **Time Range**: July 2016 – July 2018.

- **Variables**: 7 features (e.g., load, oil temperature).

- **Frequency**:

  - ETTh1/ETTh2: Hourly sampling.
  - ETTm1/ETTm2: Minute-level sampling.

- **Purpose**: Benchmark for long-term temporal dependency modeling.

### E.2 WEATHER DATASET

- **Source**: Max Planck Institute for Biogeochemistry (2020).
- **Time Range**: Full year of 2020.
- **Variables**: 21 meteorological factors (e.g., temperature, humidity).
- **Frequency**: 10-minute sampling.
- **Purpose**: Evaluate multivariate interaction modeling in complex environmental systems.

### E.3 TRAFFIC DATASET

- **Source**: 862 highway sensors in San Francisco Bay Area.
- **Time Range**: January 2015 – December 2016.
- **Variables**: Road occupancy rates (1D per sensor).
- **Frequency**: Hourly sampling.
- **Purpose**: Test spatial-temporal correlation handling in high-dimensional data.

### E.4 ELECTRICITY DATASET

- **Source**: Hourly consumption records of 321 customers.
- **Time Range**: 2012 – 2014.
- **Variables**: Electricity usage (1D per customer).
- **Frequency**: Hourly sampling.
- **Purpose**: Benchmark for high-dimensional multivariate forecasting.

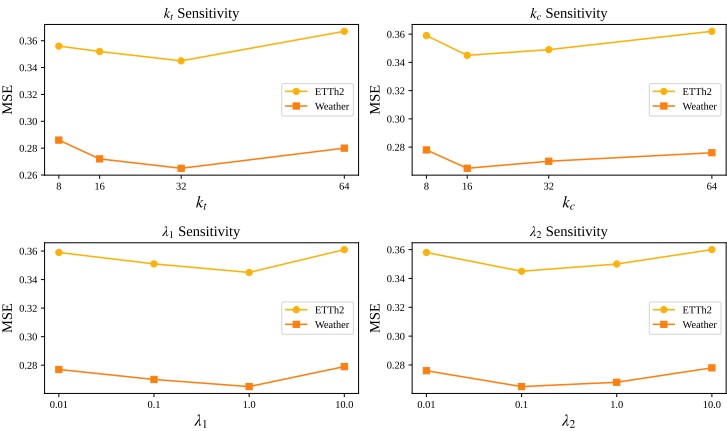

Figure 7: Sensitivity analysis of key hyperparameters on ETTh2 and Weather datasets.

#### E.4.1 HYPERPARAMETER SENSITIVITY

We conduct a hyperparameter sensitivity study on two representative datasets, ETTh2 and Weather, focusing on four key hyperparameters: the number of groups along the temporal axis ($k_t$), the number of groups along the channel axis ($k_c$), and the weights of the temporal and channel modeling losses ($\lambda_1$ and $\lambda_2$). As shown in Figure 7, MTS-UNMixers achieves the best performance when $k_t = 32$, $k_c = 16$, $\lambda_1 = 1.0$, and $\lambda_2 = 0.1$. Several trends can be observed from the results. First, with respect to the temporal grouping parameter $k_t$, the model performs best at $k_t = 32$, while both smaller values (e.g., 8) and larger values (e.g., 64) result in slight performance degradation. This suggests that overly coarse or overly fine temporal decomposition may hinder stable temporal modeling. Second, for the channel grouping parameter $k_c$, a moderate value such as 16 yields better results. On high-dimensional datasets, excessively large group sizes may lead to over-compression

of features, thereby reducing the model's expressive capacity. Regarding the loss weights $\lambda_1$ and $\lambda_2$, we find that setting $\lambda_1 = 1.0$ and $\lambda_2 = 0.1$ leads to the most balanced outcomes.

Overall, MTS-UNMixers demonstrates stable performance across a wide range of hyperparameter settings, showing good robustness and tunability. Moderate group sizes and symmetric structural weights generally yield optimal or near-optimal forecasting accuracy.

### E.5 MODEL EFFICIENCY ANALYSIS

To summarize the model performance and efficiency, we calculate relative performance rankings to compare the baselines. The ranking is based on the common models used across all five tasks: Informer, Autoformer, FEDformer, PatchTST, TimesNet, TimeXer, and our proposed MTS-UNMixers, totaling six models. We compare the models using three efficiency metrics under different input and output sequence lengths: the number of parameters (Params) and runtime (s/iter). All experiments were conducted at a unified level to ensure fairness. The results are presented in Table 4. As shown in the Table 4, significant runtime differences exist between models with the same input and output sequence lengths. Although MTS-UNMixers does not outperform PatchTST, it still demonstrates competitive performance.

Table 4: Model Efficiency Analysis.

(a) Running Time Efficiency Analysis (s/iter)

| Model | | Informer | Autoformer | FEDformer | PatchTST | TimesNet | TimeXer | MTS-UNMixers |
|---|---|---|---|---|---|---|---|---|
| | 96 | 0.0078 | 0.0109 | 0.0859 | 0.0033 | 0.0428 | 0.0059 | 0.0054 |
| Future Length | 192 | 0.0095 | 0.0111 | 0.0860 | 0.0033 | 0.0438 | 0.0050 | 0.0054 |
| | 336 | 0.0098 | 0.0111 | 0.0864 | 0.0033 | 0.0528 | 0.0056 | 0.0054 |
| | 720 | 0.0102 | 0.0111 | 0.0867 | 0.0033 | 0.0754 | 0.0056 | 0.0054 |
| | 96 | 0.0078 | 0.0109 | 0.0859 | 0.0036 | 0.0428 | 0.0059 | 0.0054 |
| History Length | 192 | 0.0079 | 0.0103 | 0.0777 | 0.0036 | 0.0408 | 0.0061 | 0.0054 |
| | 336 | 0.0080 | 0.0104 | 0.0669 | 0.0036 | 0.0588 | 0.0061 | 0.0054 |
| | 720 | 0.0082 | 0.0104 | 0.0669 | 0.0036 | 0.0791 | 0.0049 | 0.0054 |

(b) Model Parameter Efficiency Analysis (M)

| Model | | Informer | Autoformer | FEDformer | PatchTST | TimesNet | TimeXer | MTS-UNMixers |
|---|---|---|---|---|---|---|---|---|
| | 96 | 11.33 | 10.54 | 16.12 | 7.49 | 37.53 | 0.15 | 3.32 |
| Future Length | 192 | 11.33 | 10.54 | 16.12 | 7.49 | 37.53 | 0.16 | 3.38 |
| | 336 | 11.33 | 10.54 | 16.12 | 7.49 | 37.53 | 0.18 | 3.99 |
| | 720 | 11.33 | 10.54 | 16.12 | 7.49 | 37.53 | 0.22 | 4.34 |
| | 96 | 11.33 | 10.54 | 16.12 | 7.49 | 37.53 | 0.15 | 3.32 |
| History Length | 192 | 11.33 | 10.54 | 16.12 | 7.49 | 37.53 | 0.16 | 4.27 |
| | 336 | 11.33 | 10.54 | 16.12 | 7.49 | 37.53 | 0.18 | 5.76 |
| | 720 | 11.33 | 10.54 | 16.12 | 7.49 | 37.53 | 0.22 | 10.18 |

## F  VARYING LOOKBACK WINDOW

The results in Table 5 illustrate the impact of varying history lengths on the prediction performance of MTS-UNMixers across different prediction steps (96, 192, 336, and 720) on the ETTh1, ETTm1, and Weather datasets. Overall, longer history lengths contribute to improved prediction accuracy, particularly for longer prediction steps, highlighting the advantages of utilizing extended historical information. Theoretically, prediction performance should increase as the input history sequence length grows.

For the ETTh1, ETTm1, and Weather datasets, extending the history length significantly enhances the prediction accuracy of MTS-UNMixers, with especially strong improvements for long prediction steps. For instance, increasing the history length from 96 to 720 across different datasets yields substantial improvements in both MSE and MAE, indicating that a longer history window helps the model better capture long-term dependencies and periodic features in time series data. In summary, the prediction performance of MTS-UNMixers consistently improves with extended history lengths, particularly in tasks involving longer prediction steps. These experimental results demonstrate that

extending the historical sequence length effectively enhances the model's forecasting capability, underscoring the importance of capturing long-term dependencies in time series forecasting.

We also conducted a visual comparison with other models, as shown in Fig. 8. It can be seen that our results remain ahead, with a steady decrease in MSE and a gradual improvement in performance.

Table 5: Prediction performance of MTS-UNMixers under different history lengths (96, 192, 336, 720). The best performance is highlighted in bold.

| History Length | | 96 | | 192 | | 336 | | 720 | |
|---|---|---|---|---|---|---|---|---|---|
| Metric | | MSE | MAE | MSE | MAE | MSE | MAE | MSE | MAE |
| ETTh1 | 96 | 0.368 | 0.388 | 0.361 | 0.379 | 0.359 | 0.377 | **0.355** | **0.371** |
| | 192 | 0.427 | 0.419 | 0.424 | 0.413 | 0.419 | 0.409 | **0.416** | **0.412** |
| | 336 | 0.443 | 0.433 | 0.441 | 0.431 | 0.438 | 0.431 | **0.432** | **0.428** |
| | 720 | 0.454 | 0.429 | 0.448 | 0.427 | 0.444 | 0.414 | **0.439** | **0.409** |
| ETTm1 | 96 | 0.317 | 0.350 | 0.308 | 0.340 | 0.291 | 0.339 | **0.291** | **0.326** |
| | 192 | 0.360 | 0.378 | 0.351 | 0.374 | **0.343** | 0.369 | 0.346 | **0.355** |
| | 336 | 0.388 | 0.400 | 0.376 | 0.396 | **0.365** | 0.386 | 0.371 | **0.369** |
| | 720 | 0.454 | 0.429 | 0.449 | 0.422 | 0.425 | 0.419 | **0.414** | **0.417** |
| Weather | 96 | 0.158 | 0.195 | 0.158 | 0.191 | 0.153 | 0.189 | **0.149** | **0.183** |
| | 192 | 0.201 | 0.237 | 0.200 | 0.240 | 0.194 | 0.239 | **0.193** | **0.235** |
| | 336 | 0.254 | 0.286 | 0.254 | 0.279 | 0.248 | 0.274 | **0.243** | **0.271** |
| | 720 | 0.339 | 0.336 | 0.335 | 0.329 | 0.328 | 0.328 | **0.316** | **0.324** |

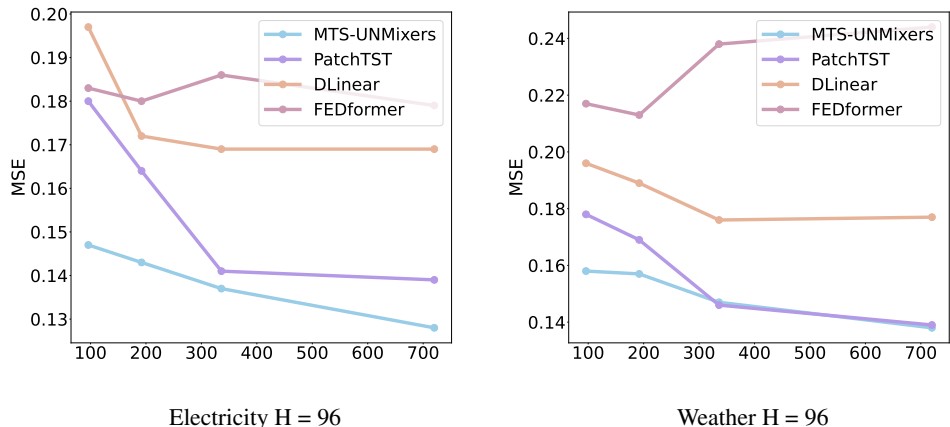

Electricity H = 96            Weather H = 96

Figure 8: The predictive performance of different models for the future sequence $H = 96$ under varying history lengths (96, 192, 336, 720).

## G  COMPLETE FORECASTING RESULTS

Table 6 summarizes the full results of multivariate time series forecasting across seven datasets (ETTh1, ETTh2, ETTm1, ETTm2, Weather, Traffic, Electricity) and four prediction lengths (96, 192, 336, 720). As summarized in Table 6, MTS-UNMixers consistently achieves top-2 performance across all seven benchmark datasets, with a total of 56 first-place results out of all evaluated settings. In particular, on the ETTh1 dataset, our model achieves significant improvements of 5.23% in MSE and 5.49% in MAE over TimeMixer, which is the strongest baseline in this scenario. Similarly, on the ETTm2 dataset, MTS-UNMixers achieves a notable 5.21% reduction in MSE, indicating its robustness in handling fine-grained temporal patterns. The superior performance of MTS-UNMixers can be attributed to its ability to decouple and recombine temporal and channel-wise representations in a structured manner. By suppressing redundant components and selectively enhancing relevant patterns, the model captures cross-variable correlations more effectively while maintaining long-range temporal consistency. These results highlight the model's strong generalization capability and its adaptability to both high-dimensional and low-frequency forecasting tasks.

Table 6: Multivariate time series forecasting results. The input length T = 96. The best results are highlighted in bold, and the second-best results are underlined.

| Model | Length | ETTh1 MSE | ETTh1 MAE | ETTh2 MSE | ETTh2 MAE | ETTm1 MSE | ETTm1 MAE | ETTm2 MSE | ETTm2 MAE | Weather MSE | Weather MAE | Traffic MSE | Traffic MAE | Electricity MSE | Electricity MAE | 1st Count |
|---|---|---|---|---|---|---|---|---|---|---|---|---|---|---|---|---|
| Ours | 96 | **0.368** | **0.388** | **0.273** | **0.327** | 0.315 | **0.345** | **0.170** | **0.250** | **0.152** | **0.189** | 0.437 | **0.274** | **0.147** | **0.242** | |
| | 192 | 0.427 | **0.419** | **0.344** | **0.372** | 0.356 | 0.369 | 0.236 | 0.293 | **0.201** | **0.237** | 0.454 | **0.288** | **0.165** | 0.260 | |
| | 336 | **0.443** | **0.433** | **0.356** | **0.390** | 0.386 | 0.390 | **0.297** | **0.333** | 0.254 | 0.282 | 0.471 | 0.295 | **0.181** | **0.274** | 56 |
| | 720 | **0.454** | **0.429** | **0.405** | **0.426** | 0.442 | 0.422 | 0.392 | **0.390** | **0.339** | **0.336** | 0.504 | 0.312 | **0.210** | **0.301** | |
| | Avg. | **0.423** | **0.417** | **0.345** | **0.379** | 0.375 | **0.382** | **0.274** | **0.316** | **0.237** | **0.261** | 0.466 | **0.292** | **0.176** | **0.269** | |
| TimeXer | 96 | 0.377 | 0.397 | 0.289 | 0.340 | **0.309** | 0.352 | 0.171 | 0.255 | 0.168 | 0.209 | **0.416** | 0.280 | 0.151 | 0.247 | |
| | 192 | 0.425 | 0.426 | 0.370 | 0.391 | **0.355** | 0.378 | 0.238 | 0.300 | 0.220 | 0.254 | **0.435** | **0.288** | 0.165 | **0.261** | |
| | 336 | 0.457 | 0.441 | 0.422 | 0.434 | 0.387 | 0.399 | 0.301 | 0.340 | 0.276 | 0.294 | 0.451 | 0.295 | 0.183 | 0.280 | 12 |
| | 720 | 0.464 | 0.463 | 0.429 | 0.445 | 0.448 | 0.435 | 0.401 | 0.397 | 0.353 | 0.347 | **0.484** | 0.314 | 0.220 | 0.309 | |
| | Avg.. | 0.431 | 0.432 | 0.378 | 0.403 | 0.375 | 0.391 | 0.278 | 0.323 | 0.254 | 0.276 | **0.447** | 0.295 | 0.180 | 0.274 | |
| TimeMixer | 96 | 0.375 | 0.400 | 0.289 | 0.341 | 0.320 | 0.357 | 0.175 | 0.258 | 0.163 | 0.209 | 0.462 | 0.285 | 0.153 | 0.247 | |
| | 192 | 0.429 | 0.421 | 0.372 | 0.392 | 0.361 | 0.381 | 0.237 | 0.299 | 0.208 | 0.250 | 0.473 | 0.296 | 0.166 | 0.256 | |
| | 336 | 0.484 | 0.458 | 0.386 | 0.414 | 0.390 | 0.404 | 0.298 | 0.340 | 0.251 | 0.287 | 0.498 | 0.296 | 0.185 | 0.277 | 3 |
| | 720 | 0.498 | 0.482 | 0.412 | 0.434 | 0.454 | 0.441 | **0.391** | 0.392 | 0.339 | 0.341 | 0.506 | 0.313 | 0.225 | 0.310 | |
| | Avg. | 0.447 | 0.440 | 0.365 | 0.395 | 0.381 | 0.396 | 0.275 | 0.322 | 0.240 | 0.272 | 0.485 | 0.298 | 0.182 | 0.273 | |
| PatchTST | 96 | 0.393 | 0.408 | 0.294 | 0.343 | 0.321 | 0.360 | 0.178 | 0.260 | 0.178 | 0.219 | 0.500 | 0.315 | 0.180 | 0.259 | |
| | 192 | 0.445 | 0.434 | 0.377 | 0.393 | 0.362 | 0.384 | 0.249 | 0.307 | 0.224 | 0.259 | 0.498 | 0.299 | 0.188 | 0.268 | |
| | 336 | 0.484 | 0.451 | 0.381 | 0.409 | 0.392 | 0.402 | 0.313 | 0.346 | 0.278 | 0.298 | 0.504 | 0.319 | 0.203 | 0.288 | 0 |
| | 720 | 0.480 | 0.471 | 0.412 | 0.433 | 0.461 | 0.439 | 0.400 | 0.398 | 0.353 | 0.346 | 0.542 | 0.335 | 0.239 | 0.321 | |
| | Avg. | 0.451 | 0.441 | 0.366 | 0.395 | 0.384 | 0.396 | 0.285 | 0.328 | 0.258 | 0.281 | 0.511 | 0.317 | 0.203 | 0.284 | |
| TimesNet | 96 | 0.384 | 0.402 | 0.340 | 0.374 | 0.338 | 0.375 | 0.187 | 0.267 | 0.172 | 0.220 | 0.593 | 0.321 | 0.168 | 0.272 | |
| | 192 | 0.436 | 0.429 | 0.402 | 0.414 | 0.374 | 0.387 | 0.249 | 0.309 | 0.219 | 0.261 | 0.617 | 0.336 | 0.184 | 0.289 | |
| | 336 | 0.491 | 0.469 | 0.452 | 0.452 | 0.410 | 0.411 | 0.321 | 0.351 | 0.280 | 0.306 | 0.629 | 0.336 | 0.198 | 0.300 | 0 |
| | 720 | 0.521 | 0.500 | 0.462 | 0.468 | 0.478 | 0.450 | 0.408 | 0.403 | 0.365 | 0.359 | 0.640 | 0.350 | 0.220 | 0.320 | |
| | Avg. | 0.458 | 0.450 | 0.414 | 0.427 | 0.400 | 0.406 | 0.291 | 0.333 | 0.259 | 0.287 | 0.620 | 0.336 | 0.193 | 0.295 | |
| FITS | 96 | 0.701 | 0.558 | 0.353 | 0.387 | 0.693 | 0.548 | 0.229 | 0.307 | 0.215 | 0.271 | 1.410 | 0.805 | 0.846 | 0.762 | |
| | 192 | 0.718 | 0.570 | 0.428 | 0.429 | 0.710 | 0.557 | 0.284 | 0.337 | 0.264 | 0.305 | 1.413 | 0.806 | 0.849 | 0.761 | |
| | 336 | 0.723 | 0.581 | 0.454 | 0.455 | 0.722 | 0.566 | 0.338 | 0.369 | 0.312 | 0.336 | 1.429 | 0.809 | 0.861 | 0.765 | 0 |
| | 720 | 0.712 | 0.595 | 0.451 | 0.460 | 0.746 | 0.581 | 0.433 | 0.419 | 0.381 | 0.377 | 1.502 | 0.820 | 0.892 | 0.775 | |
| | Avg. | 0.714 | 0.576 | 0.422 | 0.433 | 0.718 | 0.563 | 0.321 | 0.358 | 0.293 | 0.322 | 1.439 | 0.810 | 0.862 | 0.766 | |
| DLinear | 96 | 0.386 | 0.432 | 0.333 | 0.476 | 0.345 | 0.372 | 0.193 | 0.292 | 0.196 | 0.255 | 0.650 | 0.396 | 0.197 | 0.282 | |
| | 192 | 0.437 | 0.459 | 0.477 | 0.541 | 0.380 | 0.389 | 0.284 | 0.362 | 0.237 | 0.296 | 0.598 | 0.370 | 0.196 | 0.285 | |
| | 336 | 0.481 | 0.516 | 0.594 | 0.657 | 0.413 | 0.413 | 0.369 | 0.427 | 0.283 | 0.335 | 0.605 | 0.373 | 0.209 | 0.301 | 0 |
| | 720 | 0.519 | 0.452 | 0.831 | 0.515 | 0.474 | 0.453 | 0.554 | 0.522 | 0.345 | 0.381 | 0.645 | 0.394 | 0.245 | 0.333 | |
| | Avg. | 0.456 | 0.465 | 0.559 | 0.547 | 0.403 | 0.407 | 0.350 | 0.401 | 0.265 | 0.317 | 0.625 | 0.383 | 0.212 | 0.300 | |
| FEDformer | 96 | 0.376 | 0.419 | 0.346 | 0.388 | 0.378 | 0.418 | 0.203 | 0.287 | 0.217 | 0.296 | 0.562 | 0.349 | 0.183 | 0.297 | |
| | 192 | **0.420** | 0.448 | 0.429 | 0.439 | 0.426 | 0.441 | 0.269 | 0.328 | 0.276 | 0.336 | 0.562 | 0.346 | 0.195 | 0.308 | |
| | 336 | 0.459 | 0.465 | 0.496 | 0.487 | 0.445 | 0.459 | 0.325 | 0.366 | 0.339 | 0.380 | 0.570 | 0.323 | 0.212 | 0.313 | 1 |
| | 720 | 0.506 | 0.507 | 0.463 | 0.474 | 0.543 | 0.490 | 0.421 | 0.415 | 0.403 | 0.428 | 0.596 | 0.368 | 0.231 | 0.343 | |
| | Avg. | 0.440 | 0.460 | 0.434 | 0.447 | 0.448 | 0.452 | 0.305 | 0.349 | 0.309 | 0.360 | 0.573 | 0.347 | 0.205 | 0.315 | |
| TiDE | 96 | 0.427 | 0.450 | 0.304 | 0.359 | 0.356 | 0.381 | 0.182 | 0.264 | 0.202 | 0.261 | 0.568 | 0.352 | 0.194 | 0.277 | |
| | 192 | 0.472 | 0.486 | 0.394 | 0.422 | 0.391 | 0.399 | 0.256 | 0.323 | 0.242 | 0.298 | 0.612 | 0.371 | 0.193 | 0.280 | |
| | 336 | 0.527 | 0.527 | 0.385 | 0.421 | 0.424 | 0.423 | 0.313 | 0.354 | 0.287 | 0.335 | 0.605 | 0.374 | 0.206 | 0.296 | 0 |
| | 720 | 0.644 | 0.605 | 0.463 | 0.475 | 0.480 | 0.456 | 0.419 | 0.410 | 0.351 | 0.386 | 0.647 | 0.410 | 0.242 | 0.328 | |
| | Avg. | 0.518 | 0.517 | 0.387 | 0.419 | 0.413 | 0.415 | 0.293 | 0.338 | 0.271 | 0.320 | 0.608 | 0.377 | 0.209 | 0.295 | |
| Stationary | 96 | 0.513 | 0.491 | 0.476 | 0.458 | 0.386 | 0.398 | 0.192 | 0.274 | 0.173 | 0.223 | 0.612 | 0.338 | 0.169 | 0.273 | |
| | 192 | 0.534 | 0.504 | 0.512 | 0.493 | 0.459 | 0.444 | 0.280 | 0.339 | 0.245 | 0.285 | 0.613 | 0.371 | 0.182 | 0.286 | |
| | 336 | 0.588 | 0.535 | 0.552 | 0.551 | 0.495 | 0.464 | 0.334 | 0.361 | 0.321 | 0.338 | 0.618 | 0.328 | 0.200 | 0.304 | 0 |
| | 720 | 0.643 | 0.616 | 0.562 | 0.560 | 0.585 | 0.516 | 0.417 | 0.413 | 0.414 | 0.410 | 0.653 | 0.355 | 0.222 | 0.321 | |
| | Avg. | 0.570 | 0.537 | 0.526 | 0.516 | 0.481 | 0.456 | 0.306 | 0.347 | 0.288 | 0.314 | 0.624 | 0.340 | 0.193 | 0.296 | |
| Autoformer | 96 | 0.449 | 0.459 | 0.346 | 0.388 | 0.505 | 0.475 | 0.255 | 0.339 | 0.266 | 0.223 | 0.613 | 0.338 | 0.201 | 0.317 | |
| | 192 | 0.500 | 0.482 | 0.456 | 0.452 | 0.553 | 0.496 | 0.281 | 0.340 | 0.307 | 0.285 | 0.616 | 0.340 | 0.222 | 0.334 | |
| | 336 | 0.521 | 0.496 | 0.482 | 0.486 | 0.621 | 0.537 | 0.339 | 0.372 | 0.359 | 0.338 | 0.622 | 0.328 | 0.231 | 0.338 | 0 |
| | 720 | 0.514 | 0.512 | 0.515 | 0.511 | 0.671 | 0.561 | 0.433 | 0.432 | 0.419 | 0.410 | 0.660 | 0.355 | 0.254 | 0.361 | |
| | Avg. | 0.496 | 0.487 | 0.450 | 0.459 | 0.588 | 0.517 | 0.327 | 0.371 | 0.338 | 0.314 | 0.628 | 0.340 | 0.227 | 0.338 | |

# H    PREJECTION VISUALIZATION

To provide an intuitive understanding of the forecasting process, we present the prediction results from the electricity dataset in Fig. 9. The results of five models are recorded for a 96-input, 336-prediction setting. From the figure, we can observe that our model performs relatively well during cyclical fluctuations. As shown in the Fig. 9, MTS-UNMixers can respond accurately to fluctuations in signal generation cycles and trends, with precise predictions, especially in the areas highlighted by the green boxes.

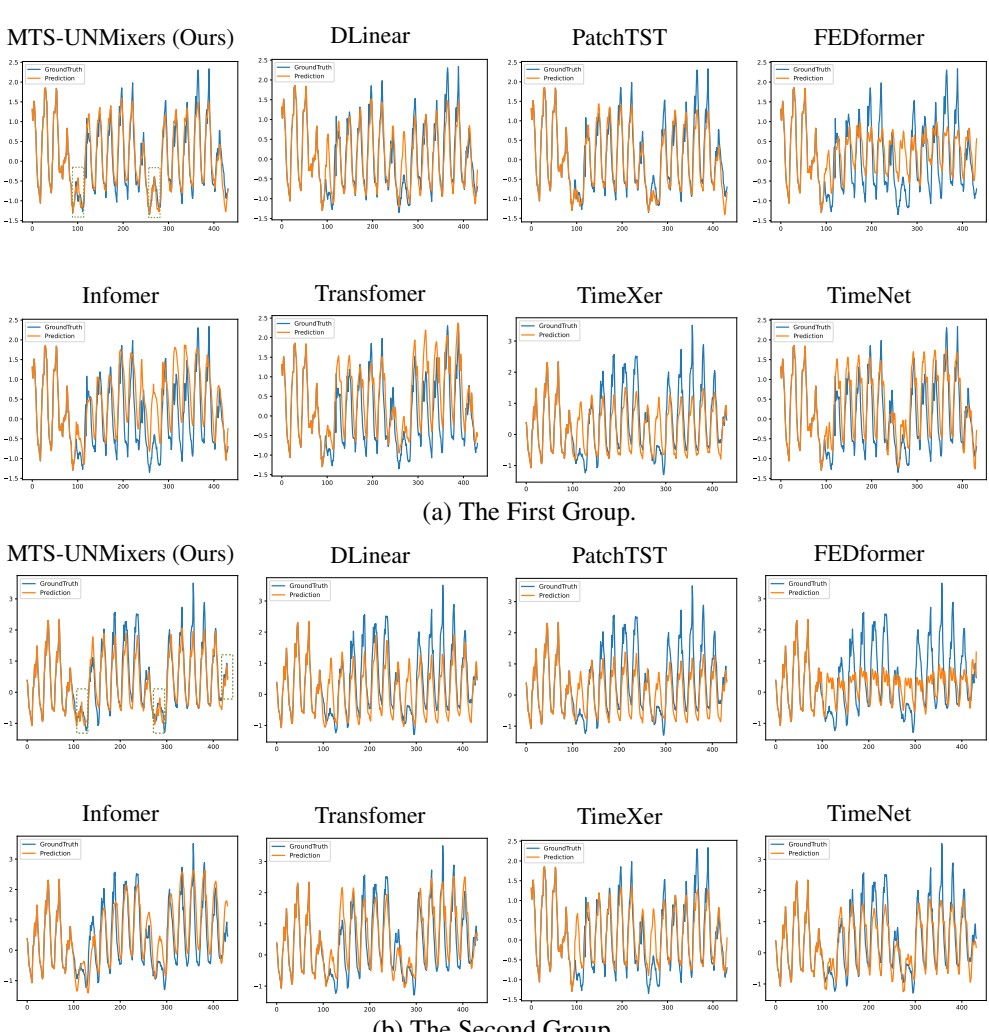

(a) The First Group.

(b) The Second Group.

Figure 9: Visualization of MTS-UNMixers' prediction results on Electricity. The input length $T = 96$. The results of two experimental groups are provided.

