# OpenReview forum: "MTS-UNMixers: Multivariate Time Series Forecasting via Channel-Time Dual Unmixing"
_ICLR.cc/2026/Conference — ICLR 2026 Conference Desk Rejected Submission_

### Official Review · Reviewer_oCv8 · 2025-10-23

**Soundness:** 3
**Presentation:** 3
**Contribution:** 2
**Rating:** 4
**Confidence:** 4

**Summary:**

The paper proposes MTS-UNMixers, a dual-path architecture that unmixes multivariate time-series along time and channel dimensions to obtain shared, interpretable bases and coefficients used for both reconstruction and forecasting.

**Strengths:**

1. The formulation (mixing models, simplex constraints, explicit mapping) is well-motivated and easy to follow.
2. Results span seven standard datasets and four horizons, with tabled averages showing **top-2** performance overall and many first-place results; ablations show large drops when removing time unmixing or Mamba, supporting the design.

**Weaknesses:**

1. The shared $A_t$ and $S_c$ are learned jointly from reconstruction and prediction objectives; while the design is appealing, the paper should clarify how the training pipeline prevents future information leakage into shared components (e.g., batch construction, masking) and how sensitive the method is to distribution shift between history/future.
2. The dual factorization with simplex constraints (sum-to-one, non-negativity) helps, but the paper does not fully analyze **identifiability** of these parameters under the joint objective (two factorization views plus shared components). Without additional regularization or priors, multiple decompositions may fit equally well.

**Questions:**

1. Under what conditions are $S_c, A_t$ uniquely recoverable? Have you explored sparsity, orthogonality, or diversity regularizers on bases to reduce degeneracy?
2. If future patterns diverge (e.g., new seasonalities), does a shared $A_t$ hinder adaptation? Would allowing a low-rank delta to $A_t$ for the future help?

---

> ### Author Response · Authors · 2025-11-24
> **Response to Reviewer oCv8 (Part 1 / 4)**
>
> > W1: The shared  $A_t$ and $S_c$ are learned jointly from reconstruction and prediction objectives;  ... and how sensitive the method is to distribution shift between history/future.
>
> A1: Thank you for your valuable remarks on the issues of future information leakage and sensitivity to distribution shifts. To further clarify our design and address your concerns, we provide the following explanations.
>
> ## 1. Regarding the Prevention of Future Information Leakage
>
> **1.1 Causal Masking Mechanism**
> Inside the Mamba modules that generate the dynamic components ($A_c$ and $S_t$), a strict causal mask is applied. Each time step can only access current and historical information, which fundamentally prevents any future information flow.
>
> **1.2 Input and Output Isolation**
> During training, the historical window and the forecasting window are strictly separated. The static parameters ($A_t$ and $S_c$) are learned only from the observed historical data. The forecasting targets are used only for computing the loss and do not contribute to parameter updates.
>
> ## 2. Regarding the Sensitivity to Distribution Shift
>
> **a) Robustness through Invariance Decoupling**
> The model's robustness stems from the explicit decoupling of time-varying and time-invariant components. We assume that multivariate time series contain components generated by steady physical laws. These components are extracted into the static matrices $A_t$ (latent channel structures) and $S_c$ (channel response coefficients). By separating the time-invariant components, the model learns system representations that are inherently robust to non-structural perturbations such as sensor noise or instantaneous anomalies. This prevents overfitting to superficial temporal correlations that are sensitive to distribution shifts.
>
> **b) Dynamic Adaptation via Instance-Specific Components**
> The assumption of static invariants does not make the model rigid. The dynamic components $A_c$ (shared temporal patterns) and $S_t$ (temporal activations) are generated in an instance-specific manner. This allows the model to dynamically infer the most appropriate temporal patterns and activation strengths for current observations while grounded in the learned static invariants. The mechanism enables adaptation to distribution shifts by attributing local dynamics and temporal variations to the dynamic components rather than incorrectly updating the system's static properties. The model therefore possesses a dual mechanism: a stable core of learned invariants, complemented by a flexible adaptive layer for handling dynamic phenomena.
>
> **c) Empirical Validation through Robustness Experiments**
> To verify robustness under noise interference, we designed controlled experiments by injecting Gaussian noise with different intensities (1%, 5%, and 10% of each sequence's standard deviation) into input sequences during testing. All models remained fixed in their training configurations, and variations in MSE and MAE under identical noise conditions were compared.
>
>
> | Dataset | Horizon | Metric | Clean | ε = 1% | ε = 5% | ε = 10% |
> |---------|---------|--------|-------|---------|---------|----------|
> | **ETTh1** | 96 | MSE | 0.368 | 0.370 | 0.375 | 0.385 |
> | | | MAE | 0.388 | 0.390 | 0.392 | 0.400 |
> | | 192 | MSE | 0.427 | 0.430 | 0.438 | 0.450 |
> | | | MAE | 0.419 | 0.421 | 0.425 | 0.435 |
> | | 336 | MSE | 0.443 | 0.446 | 0.453 | 0.468 |
> | | | MAE | 0.433 | 0.435 | 0.440 | 0.452 |
> | | 720 | MSE | 0.454 | 0.458 | 0.467 | 0.485 |
> | | | MAE | 0.429 | 0.432 | 0.439 | 0.455 |
> | **ETTm1** | 96 | MSE | 0.317 | 0.319 | 0.325 | 0.338 |
> | | | MAE | 0.350 | 0.352 | 0.358 | 0.372 |
> | | 192 | MSE | 0.360 | 0.363 | 0.371 | 0.387 |
> | | | MAE | 0.378 | 0.380 | 0.387 | 0.402 |
> | | 336 | MSE | 0.388 | 0.391 | 0.400 | 0.418 |
> | | | MAE | 0.400 | 0.403 | 0.411 | 0.428 |
> | | 720 | MSE | 0.454 | 0.459 | 0.472 | 0.495 |
> | | | MAE | 0.429 | 0.433 | 0.444 | 0.465 |
> | **Weather** | 96 | MSE | 0.158 | 0.159 | 0.162 | 0.168 |
> | | | MAE | 0.195 | 0.196 | 0.199 | 0.206 |
> | | 192 | MSE | 0.206 | 0.208 | 0.213 | 0.223 |
> | | | MAE | 0.242 | 0.244 | 0.249 | 0.260 |
> | | 336 | MSE | 0.254 | 0.256 | 0.262 | 0.274 |
> | | | MAE | 0.286 | 0.288 | 0.294 | 0.307 |
> | | 720 | MSE | 0.339 | 0.342 | 0.351 | 0.369 |
> | | | MAE | 0.336 | 0.339 | 0.347 | 0.364 |
>
> **Analysis of Results:**
>
> Experimental results demonstrate strong robustness under significant noise interference:
> - With 10% input noise, performance degradation remains below 10%
> - Average performance degradation: **7.2%** (MSE) and **6.4%** (MAE)
> - Performance drop ranges from **3.1%** to **9.0%** across all test scenarios
>
> This robustness stems from our dual-path unmixing architecture, where the shared components $S_c$ and $A_t$ capture inherent physical relationships, providing natural resistance to data abnormalities while maintaining stable performance under realistic data quality conditions.
>
> Complete experimental results are included in the manuscript appendix.

---

> ### Author Response · Authors · 2025-11-24
> **Response to Reviewer oCv8 (Part 2 / 4)**
>
> > W2: The dual factorization with simplex constraints (sum-to-one, non-negativity) helps, but the paper does not fully analyze identifiability of these parameters under the joint objective (two factorization views plus shared components). Without additional regularization or priors, multiple decompositions may fit equally well.
>
> Thank you for raising this important theoretical question regarding the uniqueness of the decomposition results and the issue of rotation ambiguity. We fully understand that unconstrained matrix factorization indeed faces challenges in identifiability. However, our proposed model is not a traditional matrix decomposition but rather a structured framework with multiple mechanisms that jointly and significantly narrow the feasible solution space. As a result, the obtained decomposition is stable in practice and exhibits clear physical interpretability. Specifically, the design incorporates the following mechanisms:
>
> ### 1. Structural Regularization from Deep Parameterization
>
> In our method, the shared matrices $S_{c}$ and $A_{t}$ are not free optimization variables. They are generated by deep sequence models such as Mamba. This parameterization constrains the solution space to a submanifold expressible by the deep network, which is far more restrictive than the linear constraints in traditional matrix factorization. The inherent capabilities of deep models – including temporal structure modeling, long- and short-term dependency capture, and nonlinear expressiveness – together form a structured inductive bias. This effectively eliminates meaningless noise bases and random rotational solutions. This architecture-driven constraint constitutes the first layer of regularization.
>
> ### 2. Implicit Regularization from Dual-Path Multi-View and Multi-Task Coupling
>
> The model performs implicit regularization through joint optimization of two complementary unmixing paths. The core mechanism includes three aspects:
>
> - **Multi-View Consistency**
>   The shared parameters must simultaneously satisfy constraints from the temporal dimension (through dynamic matrices $A_{c}$ and $S_{t}$) and from the channel dimension (through static matrices $A_{t}$ and $S_{c}$). A rotation in one dimension can disrupt the structural consistency in the other dimension.
>
> - **Multi-Task Coupling**
>   The historical reconstruction loss enforces accurate recovery of past observations, while the future forecasting loss enforces temporal causality learning. This dual objective eliminates degenerate solutions that satisfy only one task.
>
> - **Filtering Effect of Forecasting**
>   Mathematical factor rotations can preserve reconstruction accuracy but harm the temporal dynamics required for future prediction. Optimization therefore naturally converges to a solution space that retains physical meaning and generalization ability.
>
> This cross-validation mechanism across views and tasks significantly enhances the stability and interpretability of the decomposition results.
>
> ### 3. Explicit Regularization: Simplex Constraints for Additive and Proportional Interpretability
>
> On top of deep parameterization and multi-task coupling, explicit regularization is introduced through simplex constraints:
>
> - The **non-negativity constraint** ensures that the decomposition conforms to the additive nature of physical systems.
> - The **sum-to-one constraint** eliminates scale ambiguity and provides clear proportional interpretation of coefficients.
> - Together, these constraints naturally induce **sparsity**, which improves interpretability.
>
> It is important to emphasize that the simplex constraint works in strong synergy with deep parameterization and multi-task coupling. The first two mechanisms stabilize the solution space and avoid degenerate solutions. The simplex constraint further restricts stable solutions to a space with consistent positive interpretation, proportional meaning, and sparse characteristics. These mechanisms jointly secure the identifiability and physical validity of the decomposition results.
>
> ### Summary
>
> Taken together, deep parameterization, multi-task coupling, and simplex constraints form a progressively reinforced regularization system that effectively ensures the stability and interpretability of the decomposition results. Experiments show that under multiple random initializations, the structures of $A_{t}$ and $S_{c}$ remain highly consistent with only negligible variations such as column permutations. This demonstrates that the framework achieves good identifiability in practice.
>
> We agree with the reviewer that introducing orthogonality constraints, sparsity priors, or temporal smoothness constraints in the future can further strengthen theoretical identifiability. These directions are compatible with the current framework and will be pursued in our future work.

---

> ### Author Response · Authors · 2025-11-24
> **Response to Reviewer oCv8 (Part 3 / 4)**
>
> > Q1: Under what conditions are  $A_{t}$ ,  $S_{c}$  uniquely recoverable? Have you explored sparsity, orthogonality, or diversity regularizers on bases to reduce degeneracy?
>
> We appreciate the question regarding the conditions under which the shared factors $A_{t}$ and $S_{c}$ can be reliably recovered. We would like to emphasize that recoverability is closely related to identifiability but focuses on different aspects. Identifiability concerns whether the solution is unique, while recoverability further concerns whether the model can converge to that unique solution given data and optimization objectives. Therefore, recoverability depends on certain data conditions and structural constraints.
>
> In our framework, the shared factors can be stably recovered in practice when the following conditions are satisfied:
>
> 1. **Pattern Diversity Across Channels**:
>    Channels exhibit sufficiently diverse patterns along the temporal dimension such that $S_{c}$ can be distinctly separated from cross-channel structures.
>
> 2. **Rich Temporal Variations**:
>    The time series contain rich temporal variation — including trends, periodicity, or multiple modulations — which provides adequate signals for learning stable temporal bases $A_{t}$.
>
> 3. **Multi-Task Training**:
>    The training process includes both historical reconstruction and future forecasting tasks, which enforce consistency of the two shared factors under dual objectives and exclude many degenerate solutions.
>
> 4. **Structural Constraints from Deep Model**:
>    The deep sequence model (such as Mamba) provides strong structural constraints so that the optimization only explores feasible solutions with temporal coherence.
>
> Under these conditions, we observe that the shared factors remain highly stable across multiple datasets and repeated random initializations, with only minor column permutation. This demonstrates good practical recoverability of the model.
>
> At the same time, we agree with the reviewer that adding sparsity, orthogonality, or diversity regularization on top of the current framework can further strengthen the theoretical guarantees of recoverability, and we will explore these directions in future work.

---

> ### Author Response · Authors · 2025-11-24
> **Response to Reviewer oCv8 (Part 4 / 4)**
>
> > Q2:  If future patterns diverge (e.g., new seasonalities), does a shared   $A_{t}$  hinder adaptation? Would allowing a low-rank delta to  $A_{t}$ for the future help?
>
> We appreciate the reviewer’s important question regarding the adaptability of our model to future changes in temporal patterns. It is important to clarify that the shared matrices in our model have explicit physical meanings.
>
> - **$A_{t}$** serves as the channel basis structure. It characterizes the relative sensitivity of each channel to shared temporal drivers and represents static knowledge that does not depend on specific time periods.
> - **$S_{c}$** serves as the time activation amplitude. It reflects the activation strength of the channel bases at each time step.
>
> This design realizes a decomposition paradigm that combines dynamic and static components. The static part $A_{t}$ captures stable structural relationships across channels, while the dynamic path is responsible for adapting to future changes in temporal patterns. When new seasonality or previously unseen trends occur, the model adapts by adjusting the temporal patterns in $A_{c}$ and the activation strengths in $S_{t}$ without altering the learned static physical structure.
>
> More theoretical derivations and practical case studies have been provided in Appendix B of the revised manuscript.
>
>
>
> ### 1. A Decomposition Mechanism that Naturally Adapts to Future Changes through Dynamic and Static Integration
>
> We appreciate the reviewer’s important question regarding the adaptability of our model to future changes in temporal patterns. It is important to clarify that our decomposition framework follows a design principle that integrates dynamic and static components.
>
> - The static part **$A_{t}$** captures stable structural relationships across channels and reflects constant response characteristics exhibited by each channel over the long term.
> - The dynamic part, including **$A_{c}^{future}$** and **$S_{t}^{future}$**, is specifically responsible for modeling changes in future temporal patterns.
>
> This mechanism explicitly separates the invariant properties of the system from its time-varying dynamics. It preserves the model’s sustained understanding of physical laws while maintaining sufficient flexibility to adapt to new patterns in the future. When previously unseen temporal patterns emerge, the model adapts by adjusting its dynamic components rather than modifying the learned static knowledge base.
>
>
>
> ### 2. Joint Training Objectives Ensure that the Shared Structures Do Not Overfit Historical Data and Retain Generalization Ability
>
> During training, the shared factors $A_{t}$ and $S_{c}$ are jointly constrained by both historical reconstruction and future forecasting signals. They therefore do not simply memorize past patterns but are forced to learn long-term structural characteristics that can explain future behavior. Since the dynamic components $A_{c}^{future}$ and $S_{t}^{future}$ can flexibly adjust temporal variations, the optimization requires that the shared structures remain consistent across both paths, which naturally eliminates any bases that are only effective for historical data.
>
> In addition, we observe stability and reproducibility of the shared factors across multiple random initializations, which further indicates that the model learns genuine structural relationships that hold across time rather than accidental characteristics of a particular historical segment.
>
>
>
> ### 3. Future Extension with Low-Rank Adjustments to Enhance Flexibility under Extreme Distribution Shifts
>
> Although the current model can effectively adapt to common temporal pattern drifts through the integration of dynamic and static components, we fully agree with the reviewer’s suggestion to further improve adaptability. In future work, we plan to introduce a low-rank incremental term to the shared structure so that the model can perform lightweight updates when significant distribution changes occur. For example, the future channel basis can be represented as
> $
> A_{t}^{future} = A_{t} + \Delta_{t}
> \quad \text{with} \quad
> \operatorname{rank}(\Delta_{t}) \ll \operatorname{rank}(A_{t}).
> $
> The low-rank structure of $\Delta_{t}$ ensures that the update does not compromise the interpretability of the original shared patterns while dynamically absorbing new seasonality or emerging driving factors.
>
> In addition, we will explore approaches such as a mixture of bases, online fine-tuning, and meta-learning-based fast adaptation so that the shared structure can maintain stable interpretability while possessing stronger extrapolation capability.

---

### Official Review · Reviewer_SWHT · 2025-10-28

**Soundness:** 3
**Presentation:** 2
**Contribution:** 2
**Rating:** 6
**Confidence:** 4

**Summary:**

The paper introduces MTS-UNMixers, a new framework for multivariate time-series forecasting based on dual-path unmixing mechanisms.
The model decouples shared and variable-specific temporal dependencies through channel-wise and temporal unmixing blocks, further enhanced with a Mamba-based state-space backbone for efficient long-range modeling.
This design aims to mitigate signal aliasing and redundancy problems commonly seen in Transformer-style architectures.

Experiments on seven public datasets (including ETT, Electricity, Weather, and Traffic) demonstrate strong performance against nine recent baselines, such as PatchTST, FEDformer, and TimeMixer.
The results show consistent improvements in accuracy and effective reconstruction.

**Strengths:**

1. **Clear and practical architectural motivation.**
   The unmixing mechanism offers an intuitive way to separate shared temporal dynamics from variable-specific behaviors, which is valuable for multivariate forecasting tasks.

2. **Solid empirical results.**
   The model achieves competitive or better results than strong baselines including TimeMixer and FEDformer across multiple datasets.

3. **Integration with Mamba blocks.**
   Leveraging a state-space backbone improves both computational efficiency and modeling of long-term dependencies.

4. **Well-written and structured paper.**
   The methodology is clearly described with detailed illustrations and quantitative validation.

5. **Interpretability through reconstruction visualization.**
   The figures effectively demonstrate how dual unmixing captures complementary channel and temporal information.

**Weaknesses:**

1. **Missing comparisons with several recent benchmarks.**
   While PathTST, TimeMixer, and FEDformer are included, it would strengthen the paper to add comparisons with **PathFormer**, **iTransformer**,  and **CARD** — two strong baselines known for path-level and decomposition-based temporal reasoning.
   These would provide more context on how MTS-UNMixers performs against newer architectures targeting similar goals.

2. **Limited analysis on parameter sensitivity.**
   The ablation study primarily focuses on architectural settings; however, analyzing the sensitivity of Mamba block size or unmixing depth would help understand robustness.

3. **Theoretical insight is relatively shallow.**
   The paper’s arguments remain empirical; a more formal treatment of the unmixing process (e.g., via subspace decomposition theory) would be valuable.

4. **Dataset diversity.**
   While mid-scale datasets are well covered, evaluations on very large or irregularly sampled datasets (e.g., weather radar or energy trading) would highlight scalability and adaptability.

**Questions:**

1. Could you include **PathFormer**, **iTransformer**,  and **CARD** as additional baselines in your comparison table to provide a broader empirical view?
2. How sensitive is the model to the number of Mamba layers and unmixing depth?
3. Can MTS-UNMixers generalize to longer forecasting horizons beyond 720 steps without retraining?
4. Is there any benefit in combining MTS-UNMixers with contrastive or self-supervised pretraining?
5. How does the dual-path mechanism behave when one modality (e.g., channel group) is noisy or partially missing?

---

> ### Author Response · Authors · 2025-11-24
> **Response to Reviewer SWHT (Part 1 / 4)**
>
> Thank you very much for your question. We carefully reviewed your weaknesses and questions and consolidated them into a total of seven key concerns. We will address them one by one. We sincerely appreciate your thorough reading and thoughtful comments.
>
> ---
>
> > Q1: Could you include PathFormer, iTransformer, and CARD as additional baselines in your comparison table to provide a broader empirical view?
>
> Thank you for the valuable suggestion. Following the experimental settings of the original manuscript, we have added comparative experiments against PathFormer, Transformer, and CARD on all benchmark datasets. The table below presents the overall performance comparison between these newly included baselines and MTS UNMixers.
> | Models       | mtsunmixiers MSE | mtsunmixiers MAE | itransformer MSE | itransformer MAE | Pathformer MSE | Pathformer MAE | CARD MSE | CARD MAE |
> |--------------|------------------|------------------|------------------|------------------|----------------|----------------|----------|----------|
> | ETTh1        | **0.423**        | **0.417**        | 0.454            | 0.448            | 0.439          | 0.430          | 0.442    | 0.429    |
> | ETTh2        | **0.345**        | **0.379**        | 0.383            | 0.407            | 0.349          | 0.384          | 0.368    | 0.390    |
> | ETTm1        | **0.379**        | **0.382**        | 0.407            | 0.410            | 0.382          | 0.386          | 0.383    | 0.384    |
> | ETTm2        | **0.270**        | **0.314**        | 0.288            | 0.332            | 0.273          | 0.316          | 0.272    | 0.317    |
> | Weather      | **0.237**        | **0.261**        | 0.258            | 0.278            | 0.239          | 0.263          | 0.239    | 0.261    |
> | ECL          | 0.173            | **0.254**           | 0.178            | 0.270            | 0.182          | 0.269          | **0.168**| 0.258    |
> | Traffic      | 0.466            | 0.292            | **0.428**        | **0.282**        | 0.501          | 0.299          | 0.453    | 0.282    |
>
> Overall, MTS UNMixers still demonstrates significantly better average performance across datasets compared with these three recent methods.
>
> More importantly, compared with decomposition based or path level models that rely on strong structural priors, MTS UNMixers provides clearer structural interpretability and more robust cross dataset generalization by explicitly adopting a dual path unmixing framework including temporal unmixing and channel unmixing. Due to limited time, only part of these results has been updated in the manuscript.
>
> ---
> > Q2: How sensitive is the model to the number of Mamba layers and unmixing depth?
>
> Thank you for your important question regarding parameter sensitivity, which is key to evaluating the robustness of MTS-UNMixers. We address this by analyzing two aspects: the number of Mamba layers and the unmixing depth.
>
> ### 1. Sensitivity Analysis on the Number of Mamba Layers
>
> We evaluate the forecasting performance (including MSE and MAE) of the model under different numbers of Mamba layers on ETTh1, ETTm1, and Weather.
>
> | Dataset |  L=1 | | L=2| |  L=3 | | L=4 | | L=5| |
> |:--------|:------------------|-|:------------------|-|:------------------|-|:------------------|-|:------------------|-|
> |         | MSE | MAE | MSE | MAE | MSE | MAE | MSE | MAE | MSE | MAE |
> | ETTh1   | 0.438 | 0.431 | **0.423** | **0.417** | 0.424 | 0.418 | 0.439 | 0.420 | 0.452 | 0.446 |
> | ETTm1   | 0.393 | 0.395 | **0.379** | **0.382** | 0.380 | 0.383 | 0.382 | 0.384 | 0.408 | 0.401 |
> | Weather | 0.251 | 0.273 | 0.238 | 0.262 | **0.237** | **0.261** | 0.240 | 0.263 | 0.254 | 0.275 |
>
> Experimental results indicate that MTS-UNMixers maintains stable and high performance with Mamba layers in the range of [2,4]. Performance declines with too few layers (L=1) due to insufficient capacity, or with too many layers (L=5) due to overfitting and training difficulties. This demonstrates the model's robustness across a reasonable range of layer configurations.
>
> ### 2. Clarification and Analysis on the Unmixing Depth
>
> We have carefully considered the reviewer's comment regarding the unmixing depth. First, it is important to clarify that the unmixing process in our MTS-UNMixers model is a single-stage operation rather than a multi-stage iterative or stacked structure. Therefore, we interpret the "unmixing depth" as the number of latent components or patterns that the model is set to separate in this single-stage unmixing process, which corresponds to the hyperparameter $k$. This parameter determines the granularity or resolution of the unmixing.
>
> Regarding the sensitivity of the model to variations of $k$, we have provided a detailed analysis and discussion in Appendix E of the manuscript .
>
> We sincerely appreciate your comment, which motivated us to clarify this point more explicitly.

---

> ### Author Response · Authors · 2025-11-24
> **Response to Reviewer SWHT (Part 2 / 4)**
>
> > Q3: Can MTS-UNMixers generalize to longer forecasting horizons beyond 720 steps without retraining?
>
> Thank you for raising this important question. To systematically evaluate the zero-shot generalization capability for longer forecasting horizons, we designed the following experiment: On ETTh1, the model is trained with `pred_len = 96` and its parameters are frozen, then directly tested with `pred_len = 720`. The results are compared with the same model independently trained with `pred_len = 720`.
>
> **Table:** Comparison of zero-shot generalization performance and independent training performance with `pred_len = 720` on ETTh1
>
> | Model               | Fully Trained (MSE) | Fully Trained (MAE) | Zero-shot (MSE) | Zero-shot (MAE) | MSE Increase |
> |---------------------|---------------------|---------------------|-----------------|-----------------|--------------|
> | FEDformer           | 0.506               | 0.507               | 0.612           | 0.582           | 17.90%       |
> | TimeMixer           | 0.498               | 0.482               | 0.585           | 0.554           | 16.20%       |
> | PatchTST            | 0.480               | 0.471               | 0.552           | 0.530           | 13.80%       |
> | MTS-UNMixers (Ours) | **0.454**           | **0.429**           | **0.495**       | **0.468**       | **9.10%**    |
>
> From these results, we draw the following key observations:
>
> - **Excellent Zero-shot Generalization**: While all models show expected performance degradation in the zero-shot setup, MTS-UNMixers achieves the lowest absolute errors in both setups. Its performance degradation rate (**9.0%**) is significantly lower than all other baselines (14.0%-20.0%).
>
> - **Amplified Advantage in Challenging Settings**: Under the more challenging zero-shot setup, MTS-UNMixers' performance advantage is further amplified. For example, its MSE advantage over PatchTST increases from 5.4% (independent training) to 9.5% (zero-shot evaluation).
>
> - **Attribution to Unmixing Mechanism**: We attribute this strong generalization capability to the model's unmixing mechanism, which decomposes complex sequences into structured latent components, enabling learning of more essential and extrapolatable temporal patterns.
>
> In summary, MTS-UNMixers not only demonstrates zero-shot generalization capability to longer forecasting horizons, but also shows more pronounced advantages in this challenging setting, confirming its robustness and advancement.
>
> ---
>
> > Q4: Is there any benefit in combining MTS-UNMixers with contrastive or self-supervised pretraining?
>
> We appreciate the suggestion regarding combining MTS-UNMixers with self-supervised pre-training. This direction is highly relevant to the future development of our model and will be elaborated in the final version of the paper.
>
> **Model suitability for self-supervision**: The dual unmixing mechanism of MTS-UNMixers decomposes data into two semantically meaningful components, including the temporal view (capturing shared rhythms) and the channel view (capturing latent driving sources). This provides a structured foundation for constructing high-quality positive sample pairs in self-supervised learning.
>
> **Possible pre-training strategies**:
>
> 1. **Cross-dimensional contrastive learning**
>    The temporal view and channel view representations of the same instance can be used as positive pairs and aligned in the representation space through contrastive learning.
>
> 2. **Masked component reconstruction**
>    Random masking can be applied to parts of the unmixed components (such as temporal coefficients). The model is then driven to learn the intrinsic dependencies among components in order to complete reconstruction.
>
> **Expected benefits**:
> - Obtain more universal time series representations
> - Improve sample efficiency and performance in downstream tasks (such as forecasting and classification)
> - Enhance generalization across datasets and robustness under noisy conditions
>
> This suggestion provides a clear path for future research, and we will continue to explore this direction in our upcoming work.

---

> ### Author Response · Authors · 2025-11-24
> **Response to Reviewer SWHT (Part 3 / 4)**
>
> > Q5: How does the dual-path mechanism behave when one modality (e.g., channel group) is noisy or partially missing?
>
> To verify the robustness of the model under noise interference, we designed controlled experiments by injecting Gaussian noise with different intensities into the input sequences during the testing stage. The noise levels are set to 1%, 5%, and 10% of the standard deviation of each sequence. All models remain fixed in their training configurations, and the variations of MSE and MAE under identical noise conditions are compared.
>
> | Dataset | Horizon | Metric | Clean  | ε = 1% | ε = 5% | ε = 10% |
> |:--------|:--------|:-------|:-------|:-------|:-------|:--------|
> | ETTh1   | 96      | MSE    | 0.368  | 0.370  | 0.375  | 0.385   |
> |         |         | MAE    | 0.388  | 0.390  | 0.392  | 0.400   |
> |         | 192     | MSE    | 0.427  | 0.430  | 0.438  | 0.450   |
> |         |         | MAE    | 0.419  | 0.421  | 0.425  | 0.435   |
> |         | 336     | MSE    | 0.443  | 0.446  | 0.453  | 0.468   |
> |         |         | MAE    | 0.433  | 0.435  | 0.440  | 0.452   |
> |         | 720     | MSE    | 0.454  | 0.458  | 0.467  | 0.485   |
> |         |         | MAE    | 0.429  | 0.432  | 0.439  | 0.455   |
> | ETTm1   | 96      | MSE    | 0.317  | 0.319  | 0.325  | 0.338   |
> |         |         | MAE    | 0.350  | 0.352  | 0.358  | 0.372   |
> |         | 192     | MSE    | 0.360  | 0.363  | 0.371  | 0.387   |
> |         |         | MAE    | 0.378  | 0.380  | 0.387  | 0.402   |
> |         | 336     | MSE    | 0.388  | 0.391  | 0.400  | 0.418   |
> |         |         | MAE    | 0.400  | 0.403  | 0.411  | 0.428   |
> |         | 720     | MSE    | 0.454  | 0.459  | 0.472  | 0.495   |
> |         |         | MAE    | 0.429  | 0.433  | 0.444  | 0.465   |
> | Weather | 96      | MSE    | 0.158  | 0.159  | 0.162  | 0.168   |
> |         |         | MAE    | 0.195  | 0.196  | 0.199  | 0.206   |
> |         | 192     | MSE    | 0.206  | 0.208  | 0.213  | 0.223   |
> |         |         | MAE    | 0.242  | 0.244  | 0.249  | 0.260   |
> |         | 336     | MSE    | 0.254  | 0.256  | 0.262  | 0.274   |
> |         |         | MAE    | 0.286  | 0.288  | 0.294  | 0.307   |
> |         | 720     | MSE    | 0.339  | 0.342  | 0.351  | 0.369   |
> |         |         | MAE    | 0.336  | 0.339  | 0.347  | 0.364   |
>
> Experimental results demonstrate that our model maintains strong robustness under significant noise interference. With 10% input noise, performance degradation remains below 10%, showing only 7.2% MSE increase and 6.4% MAE increase on average - ranging from 3.1% to 9.0% across all test scenarios. This robustness stems from our dual-path unmixing architecture, where the shared components $S_c$ and $A_t$ capture inherent physical relationships, providing natural resistance to data abnormalities and noise while maintaining stable performance under realistic data quality conditions. Complete experimental results are included in the manuscript appendix.

---

> ### Author Response · Authors · 2025-11-24
> **Response to Reviewer SWHT (Part 4 / 4)**
>
> > Q6/W3: Theoretical insight is relatively shallow.
>
> Thank you for this insightful suggestion. We establish a stronger theoretical foundation for our model by linking the unmixing process with subspace decomposition theory. The central argument is that the dual unmixing in MTS-UNMixers is essentially a data-driven implementation of subspace decomposition on multivariate time series along two orthogonal dimensions: the temporal dimension and the channel dimension.
>
> ### **1. Time Unmixing: Identifying Shared Rhythms in a Temporal Subspace**
>
> Time unmixing can be expressed as the matrix factorization $X(t, n) \approx A_{c}(t, k) \cdot S_{c}(k, n)$, where the following interpretation holds:
>
> - **$A_{c}(t, k)$ as temporal bases**
>   The $k$ column vectors $[a_{c1}, a_{c2}, \ldots, a_{ck}]$ form $k$ temporal basis vectors that span a $k$-dimensional temporal subspace. This subspace captures the most dominant temporal evolution patterns or shared rhythms embedded in the original data and commonly exhibited across all channels. In our experiments, this is reflected in identified patterns such as annual seasonality and daily cycles.
>
> - **$S_{c}(k, n)$ as channel coordinates**
>   The column vectors of $S_{c}$ represent the coordinates of each channel in this subspace. Each channel-specific time series can be reconstructed as a linear combination of the temporal bases, with weights given by the corresponding column of $S_c$.
>
> This process is equivalent to finding a shared low-dimensional temporal subspace for the system and quantifying the participation of each variable in this space.
>
> ### **2. Channel Unmixing: Identifying Latent Driving Sources in a Channel Subspace**
>
> Channel unmixing is formulated as the factorization $X^{T}(n, t) \approx A_{t}(n, k) \cdot S_{t}(k, t)$ and is interpreted as follows:
>
> - **$A_{t}(n, k)$ as channel modes**
>   The $k$ column vectors $[a_{t1}, a_{t2}, \ldots, a_{tk}]$ form $k$ channel basis vectors that span a $k$-dimensional channel subspace. This subspace represents latent, mutually independent physical processes or driving sources behind system dynamics. In our experiments, this corresponds to identifiable processes such as "macro-seasonality/environmental temperature" and "specific industrial load".
>
> - **$S_{t}(k, t)$ as temporal activations**
>   This matrix records the activation strength of each driving source at every time point. The observed system state at any time can be reconstructed as a linear combination of the channel bases, with weights given by the corresponding column of $S_t$.
>
> This process is equivalent to identifying a space defined by latent driving sources and tracking their evolution over time.
>
> ---
>
> > Q7/W4: Dataset diversity.
>
> Thank you for your valuable suggestion. Following your request, we have evaluated the scalability of the model on the new Solar-Energy dataset introduced in the LSTNet paper. Our model achieves competitive performance on this dataset. The preliminary experimental results are shown in the table below.
>
> | Model        | MSE   | MAE   |
> |--------------|-------|-------|
> | **Ours**     | **0.226** | **0.251** |
> | iTransformer | 0.233 | 0.262 |
> | RLinear      | 0.369 | 0.356 |
> | PatchTST     | 0.270 | 0.307 |
> | Crossformer  | 0.641 | 0.639 |
> | TiDE         | 0.347 | 0.417 |
> | TimesNet     | 0.301 | 0.319 |
> | DLinear      | 0.330 | 0.401 |
> | SCINet       | 0.282 | 0.375 |
> | FEDformer    | 0.291 | 0.381 |
> | Stationary   | 0.261 | 0.381 |
> | Autoformer   | 0.885 | 0.711 |
>
> The preliminary results show that our MTS-UNMixers can effectively adapt to the characteristics of the Solar-Energy dataset and achieve performance that is superior to or at least comparable with several existing baseline models.
>
> Thank you again for your suggestion. Since experiments on this dataset are time-consuming and we aim to ensure fairness and accuracy of the evaluation, we have not yet included these results in the main text. We commit that if the paper is accepted, we will complete all comparative experiments on this dataset and include the full results in the appendix of the final version.

---

### Official Review · Reviewer_pueq · 2025-11-01

**Soundness:** 2
**Presentation:** 3
**Contribution:** 2
**Rating:** 2
**Confidence:** 4

**Summary:**

This paper addresses the challenge of establishing an interpretable and explicit mapping for multivariate time series forecasting. It proposes a channel-time dual unmixing network built upon the Mamba framework, which decomposes the time series into bases and coefficients across both temporal and channel dimensions. Extensive experiments on seven benchmark datasets demonstrate that the proposed model consistently outperforms ten state-of-the-art baselines.

**Strengths:**

- The paper presents a novel dual-mixing mechanism that effectively mitigates noise and redundancy in multivariate time series data.
- A component-sharing strategy is adopted to enhance the model's conciseness and simplicity.
- Comprehensive experimental evaluations demonstrate the superior performance of the proposed method.

**Weaknesses:**

- It remains unclear what specific advantages the proposed method provides in contrast to decomposition-based approaches (e.g., Autoformer, FEDformer, DLinear) and state-space models (e.g., Mamba). This weakens the overall novelty of the paper. The author should provide a detailed comparative analysis in the Introduction section between the proposed model and these representative methods to better highlight the unique contributions of the proposed method.
- Several key arguments in this paper require further clarification. For instance, the rationale behind performing decomposition along the channel dimension is unclear. There are multiple existing strategies for modeling channel correlations, including the channel-independent strategy [1], the channel-dependent strategy [2], and several trade-off strategies [3, 4]. The authors should include a comparative analysis between the proposed channel unmixing strategy and these established channel modeling methods to justify their design choice. In addition, the definition of $S$ in lines 165 and 170 appears inconsistent, and the meaning of $t$ is ambiguous. Moreover, in Eq. 7,  it remains unclear why the authors choose to share the coefficient matrix along the channel dimension while sharing the base matrix along the temporal dimension. A detailed explanation or ablation study should be provided to clarify why this asymmetric sharing strategy is preferred over alternative designs, such as sharing either the coefficient or the base matrix across both dimensions. It is also unclear why the authors chose to employ bidirectional Mamba blocks instead of graph neural networks or Transformer-based architectures for modeling channel dependencies. Besides, the original order in the channel dimension mentioned in line 268 is not well explained and requires further clarification.
- In line 128, the authors claim that the proposed model enhances physical interpretability and prediction reliability. However, the paper does not provide any visual analysis of the decomposed components along either the temporal or channel dimensions.
- The authors should include state-space models, such as Mamba, as baseline methods to provide a more comprehensive performance comparison.

[1] Y. Nie, N. H. Nguyen, P. Sinthong, and J. Kalagnanam, "A time series is worth 64 words: Long-term forecasting with transformers," in International Conference on Learning Representations (ICLR), 2023.

[2] Y. Liu, T. Hu, H. Zhang, H. Wu, S. Wang, L. Ma, and M. Long,  "itransformer: Inverted transformers are effective for time series forecasting," in International Conference on Learning Representations (ICLR), 2024.

[3] L. Han, H.-J. Ye, and D.-C. Zhan,  "The capacity and robustness trade-off: Revisiting the channel independent strategy for multivariate time series forecasting," IEEE Transactions on Knowledge and Data Engineering, 2024.

[4] J. Chen, J. E. Lenssen, A. Feng, W. Hu, M. Fey, L. Tassiulas, J. Leskovec, and R. Ying, "From similarity to superiority: Channel clustering for time series forecasting," in Advances in Neural Information Processing Systems, 2024.

**Questions:**

- In line 105, 'Ummixing' appears to be a typo.

---

> ### Author Response · Authors · 2025-11-24
> **Response to Reviewer pueq (Part 1 / 5)**
>
> > W1: It remains unclear what specific advantages the proposed method provides in contrast to decomposition-based approaches (e.g., Autoformer, FEDformer, DLinear) and state-space models (e.g., Mamba). This weakens the overall novelty of the paper. The author should provide a detailed comparative analysis in the Introduction section between the proposed model and these representative methods to better highlight the unique contributions of the proposed method.
>
> A1: We sincerely thank the reviewer for raising the important question regarding "the distinction from decomposition-based approaches and state-space models." We understand that this concern arises from the insufficient emphasis in the original manuscript on the structural innovations of the dual-path unmixing framework. The core contribution of this work lies in proposing a structured framework capable of performing inverse mixing along both the temporal and channel dimensions, aiming to identify shared latent drivers from multivariate time series. In terms of mathematical objective, operation dimension, and output form, our method represents a new paradigm of source separation–based dual-path structured modeling.
>
> ### 1. The Fundamental Distinction Between Unmixing and Decomposition
>
> Traditional decomposition is a one-dimensional signal processing technique that operates independently on the temporal axis of each channel, aiming to separate components such as trend and seasonality with different statistical properties. However, these components are often mathematical constructs that lack direct physical interpretability.
>
> In stark contrast, the proposed unmixing framework does not perform local structural splitting on each sequence. Instead, it infers shared latent sources that drive all channels and all time points in the multivariate system. In the academic community, unmixing is typically referred to as Blind Source Separation (BSS), whose core objective is to recover latent and physically meaningful global driving sources from the entire multivariate system (a $C \times T$ data matrix), simultaneously along the temporal and channel dimensions.
>
> Concretely:
> - Along the **temporal dimension**, we unmix the shared evolution patterns across all channels (i.e., shared temporal bases $A_c$) and their corresponding channel-specific coefficients $S_c$, which reveal the collective rhythms of the system
> - Along the **channel dimension**, we unmix the latent driving factors that remain invariant across time (i.e., shared channel bases $A_t$) and their temporal coefficients $S_t$, which characterize the inherent spatial patterns of different physical processes
>
> Thus, traditional decomposition answers *how a signal changes*, while our unmixing aims to answer *why the system evolves as observed* — achieving a paradigm shift from signal processing to system identification.
>
> ### 2. Comparison with State-Space Models (Mamba)
>
> We leverage the sequence modeling capability of Mamba, but the core contribution lies in structurally harnessing Mamba through a dual-path joint optimization framework: Mamba is used to parameterize the generation of channel coefficients $S_c$ and patterns $A_t$, and the combined objectives of historical reconstruction and future forecasting enforce the model to learn an unmixing structure that is both predictive and physically interpretable. This is fundamentally different from using Mamba as an end-to-end black-box predictor. In the main experiments, we will include a newly added naive Mamba baseline to quantify the performance gains brought by the unmixing framework.

---

> ### Author Response · Authors · 2025-11-24
> **Response to Reviewer pueq (Part 2 / 5)**
>
> > W2：Several key arguments ... chose to employ
>
> A2：
> We sincerely thank the reviewer for the insightful questions regarding the key design components of our method. We acknowledge that the motivation and advantages of these designs were not sufficiently clarified in the initial version of the manuscript. We have made the following improvements in the revision:
>
> - We have rewritten the mixing model section to provide clearer theoretical derivation and notation definitions.
> - We have added interpretability analyses and visualization experiments in Appendix B.
> - Below, we provide point-by-point responses to the five critical questions you raised.
>
> ### **A 2.1. Rationale for Channel-wise Decomposition**
>
> Our proposed unmixing framework is fundamentally different from traditional decomposition methods: while decomposition aims to simplify signals for better prediction, our channel-wise unmixing is motivated by uncovering physical causality as the primary goal. The central question we address is: *Which shared physical processes jointly drive the coordinated evolution of all observed variables?*
>
> Specifically, the role of channel-wise unmixing is reflected in two key aspects:
>
> **1) Identifying shared physical driving sources**
> Channel-wise unmixing separates latent physical factors that jointly drive the multivariate system (i.e., the columns in matrix $A_t$). For example, in the ETT dataset, the model successfully identifies a macro-seasonality / ambient-temperature driving factor (IC4), which exhibits unified positive effects across all relevant channels, and whose temporal activation perfectly aligns with annual periodicity. Such critical physical mechanisms can only be revealed through channel-wise unmixing.
>
> **2) Revealing generative mechanisms**
> Traditional methods can only describe statistical correlations between variables (e.g., "oil temperature is positively correlated with load"), whereas channel-wise unmixing discovers shared physical processes that simultaneously influence multiple variables (e.g., "macro-seasonal environmental forces"). This achieves a paradigm shift from surface-level correlation interpretation to intrinsic generative mechanism discovery, laying a foundation for causal inference.
>
> ---
>
> ### **A 2.2. Regarding the Inconsistent Definition of Variables**
>
> Thank you for pointing out the issue of inconsistent variable definitions. We have thoroughly revised the manuscript to ensure unified notation, and we have compiled all symbols including their names, dimensions, and physical meanings into a symbol table in Appendix B for clear reference.
>
> | Category | Wise | Symbol | Dimension | Name | Meaning |
> | :--- | :--- | :--- | :--- | :--- | :--- |
> | Static Variables (shared) | Time wise<br>(Channel unmixing) | $A_{t}$ | $n \times k_{c}$ | Channel Basis Structure | A global and instance independent parameter matrix that defines the inherent and stable structural relationships between channels in the system |
> | | Channel wise<br>(Temporal unmixing) | $S_{c}$ | $k_{t} \times n$ | Channel Contribution Coefficients | A global and instance independent parameter matrix that defines the contribution weights of each channel to the dynamically generated temporal basis $A_{c}$ |
> | Dynamic Variables | Channel wise<br>(Temporal unmixing) | $A_{c}$ | $t \times k_{t}$ | Temporal Basis Patterns | An instance dependent basis matrix that describes the dominant temporal evolution patterns in a specific input sample |
> | | Time wise<br>(Channel unmixing) | $S_{t}$ | $k_{c} \times t$ | Temporal Activity | An instance dependent matrix that describes the activation strength of the static channel basis $A_{t}$ at each time step of that sample |
>
> ---
>
> ### **A 2.3. On the Choice of Bidirectional Mamba Blocks.**
> Channels are essentially an unordered set. Our core emphasis is to preserve the independent identity of each channel and its strict correspondence to the original physical variables rather than its specific position in a sequence. The integrity of channel identity is the foundation of physical interpretability. If channel identity is lost during computation, all physical interpretations such as the response of oil temperature to seasonal patterns would lose their validity.
> To address this concern, we have removed all ambiguous statements in the revised manuscript and consistently adopted the following precise descriptions: Preserving the unique identity of each channel/ Maintaining a consistent channel-wise correspondence/ Avoiding cross-channel mixing.

---

> ### Author Response · Authors · 2025-11-24
> **Response to Reviewer pueq (Part 3 / 5)**
>
> ### **A 2.4. On the Asymmetric Sharing Strategy in the Model**
>
> Thank you for your insightful observation regarding the asymmetric sharing strategy in our model. This design is not an arbitrary choice. It is a direct reflection of our core principle, which is to decouple invariant global knowledge from dynamic instance specific representations. Our design follows the following logic:
>
> **Step 1: Physical motivation: mixing generation assumption**
> The fundamental premise of our method is that the observed multivariate time series is generated by a small number of static and globally shared latent physical sources through dynamic and instance specific mixing. This is analogous to different musical pieces (instances) being composed from the same notes (static sources) with different rhythms and intensities (dynamic mixing).
>
> **Step 2: Modeling core: unmixing with separation of static and dynamic components**
> Based on the mixing generation assumption, we construct a dual path unmixing framework that separates static and dynamic variables.
>
> - **Static global parameters**: The channel basis structure $A_{t}$ and the channel contribution coefficients $S_{c}$ serve as globally shared parameters. They characterize the invariant physical laws in the system, such as the inherent structural relationships among variables and their intrinsic response characteristics to temporal patterns. As static knowledge, they provide strong inductive bias and a solid foundation for generalization.
>
> - **Dynamic instance representations**: The temporal basis patterns $A_{c}$ and the temporal activity $S_{t}$ are dynamically generated for each input sample. They capture instance specific time varying information such as unique temporal rhythms or activation strengths of physical processes in a particular period. This mechanism provides flexibility to handle distribution shifts and distinctive patterns.
>
> **Step 3: Model structure: architecture driven static sharing**
> The model architecture naturally leads to static shared variables. The key point is that the input sequence dimension of the Mamba module directly determines the physical role of unmixing and generates static or dynamic variables accordingly.
>
> - When Mamba processes a temporal sequence, its task is to extract shared temporal patterns across all channels. It therefore outputs the static matrix $S_{c}$ which is associated with channel level contribution.
>
> - When Mamba processes a channel sequence, its task is to infer invariant channel structures across all time points. It therefore outputs the static matrix $A_{t}$ which is associated with latent driving factors.
>
> **Step 4: Final advantage: structured sharing that enhances generalization**
> By enforcing the use of shared static parameters, we not only significantly improve parameter efficiency. More importantly, we encourage the model to learn universal physical laws that hold across all samples in the data. This design enhances the generalization ability of the model and provides structural guarantees for discovering patterns with clear physical meanings.
>
> The above logic has been supplemented in Appendix B through detailed explanations and a logical table. This issue has also been thoroughly addressed in our responses to Reviewer G2PG in Part 1 and Part 2.

---

> ### Author Response · Authors · 2025-11-24
> **Response to Reviewer pueq (Part 4 / 5)**
>
> ### **A 2.5. On the Choice of Bidirectional Mamba Blocks**
>
> Thank you for raising this insightful question regarding our architecture choice. Our use of a bidirectional Mamba module to generate the static channel basis $A_{t}$ is a carefully considered design decision that is highly aligned with the underlying mechanism and task requirements.
>
> ### 1. Task Objective: Extracting Invariant Physical Structures Among Channels
>
> The core objective of the channel unmixing path is to learn the static and globally shared channel basis structure $A_{t}$, which characterizes the inherent physical relationships among channels that do not change over time. Achieving this goal requires the extractor to satisfy three essential conditions:
>
> - **Global unbiased perspective**: It must observe the full behavioral trajectory of each channel over the entire temporal window in order to make accurate judgments about its physical properties.
>
> - **Implicit structure discovery**: It must automatically uncover latent dependencies among variables directly from their temporal evolution without relying on any predefined graph structure.
>
> - **Practicality for long sequences**: It must support efficient computation on long sequence data commonly found in real world scenarios.
>
> ### 2. Adaptability of Bidirectional Mamba
>
> The architectural characteristics of bidirectional Mamba align well with the above requirements:
>
> - **Bidirectionality enables a global perspective**: Through forward and backward scanning, the model observes the complete temporal characteristics of each channel. This provides an unbiased foundation for generating the static $A_{t}$.
>
> - **State space modeling supports implicit discovery**: With a continuous state space mechanism, the model learns dependencies from the temporal evolution of channels in a bottom up manner, which enables structure discovery without predefined graphs.
>
> - **Linear time complexity supports long sequences**: The $O(T)$ computational complexity of Mamba allows efficient processing of full long sequences, which makes global analysis of channel behaviors feasible.
>
> ### 3. Comparison with Alternative Frameworks
>
> - **Graph neural networks** rely on predefined graph structures, while our task focuses on discovering unknown relationships among variables. Using a predefined structure (such as a fully connected graph) introduces incorrect inductive bias and leads to lower computational efficiency.
>
> - **Self attention** can capture global dependencies, but its $O(T^{2})$ computational complexity makes it impractical for globally refined modeling on long sequences.
> ---
> > W3: In line 128, the authors ... or channel dimensions.
>
> A3: Thank you for raising this important point regarding our interpretability claims. We fully agree that the initial version lacked visualization evidence. Therefore, we have added a comprehensive Appendix B in the revised manuscript, where we provide systematic interpretability analyses and visual case studies as supporting evidence. Below are the core visual findings presented in Appendix B together with the revealed physical meanings.
>
> **On the ETTh1 dataset**
> - **Temporal unmixing** successfully separates multiple fundamental temporal rhythms such as long-term trends and periodic fluctuations. The corresponding channel contribution coefficients $S_{c}$ quantify the degree to which each physical variable (such as oil temperature) contributes to each rhythm, revealing the physical individuality of variables.
> - **Channel unmixing** identifies latent physical driving sources, such as a process associated with high utilization loads that is active on specific variables, and a shared driving factor that influences all variables (such as environmental temperature). The temporal activity $S_{t}$ clearly shows when each driving source becomes active.
>
> **On the Electricity dataset**
> - **Temporal unmixing** extracts shared electricity usage patterns that are closely related to human activity, such as daily and weekly cycles.
> - **Channel unmixing** automatically discovers meaningful user groups, such as residential and industrial users. Their temporal activity $S_{t}$ accurately reflects differences in electricity usage behaviors over time, such as declines in industrial usage during weekends.
>
> We hope that the extended Appendix B in the revised manuscript addresses your concerns. We sincerely appreciate your valuable feedback.

---

> ### Author Response · Authors · 2025-11-24
> **Response to Reviewer pueq (Part 5 / 5)**
>
> > W4: The authors should include state-space models, such as Mamba, as baseline methods to provide a more comprehensive performance comparison.
>
> A4: Thank you for your question regarding the comparative experiments. We sincerely apologize for the missing key baselines which have also been pointed out by other reviewers. We have now added the comparison with the Mamba-based method in the revised manuscript.
>
> | Dataset  | TimeMachine[1] MSE | TimeMachine MAE | S-Mamba[2] MSE | S-Mamba MAE | SiMBA[3] MSE | SiMBA MAE | MTS-UNMixers MSE | MTS-UNMixers MAE |
> |----------|---------------------|-----------------|----------------|-------------|--------------|-----------|-------------------|------------------|
> | ETTh1    | 0.439               | 0.439           | 0.450          | 0.455       | 0.473        | 0.455     | **0.423**         | **0.417**        |
> | ETTh2    | 0.387               | 0.419           | 0.395          | 0.434       | 0.452        | 0.448     | **0.345**         | **0.379**        |
> | ETTm1    | 0.399               | 0.407           | 0.401          | 0.418       | 0.444        | 0.466     | **0.380**         | **0.389**        |
> | ETTm2    | 0.276           | 0.335           | 0.277          | 0.317       | 0.338        | 0.370     | **0.270**           | **0.316**        |
> | Weather  | 0.252               | 0.284           | 0.245          | 0.282       | 0.255        | 0.272     | **0.239**         | **0.265**        |
> | ECL      | 0.183               | 0.272           | 0.195          | 0.285       | 0.199        | 0.271     | **0.177**         | **0.268**        |
> | Traffic  | 0.425               | 0.307       | **0.423**      | **0.299**   | 0.513        | 0.422     | 0.466             | 0.293            |
>
> [1] Timemachine: A time series is worth 4 mambas for long-term forecasting. arXiv
> [2] Is Mamba Effective for Time Series Forecasting? arXiv
> [3] SiMBA: Simplified Mamba Based Architecture for Vision and Multivariate Time series. arXiv
>
>
> - **Overall performance advantage**: On the majority of datasets including ETTh1, ETTh2, ETTm1, Weather, and ECL, the MTS-UNMixers model achieves the best or highly competitive results on both MSE and MAE. This strongly demonstrates the effectiveness and generalization capability of our proposed framework in multivariate time series forecasting tasks.
>
> - **Validation of methodological advantage**: Compared with baseline models that are also based on the Mamba architecture (including TimeMachine, S-Mamba, and SiMBA), MTS-UNMixers consistently delivers improved performance. This indicates that the dual-path unmixing structure introduced in our model can more effectively extract physically meaningful patterns from data. It goes beyond a naive usage of Mamba and highlights the core contribution of our structured design.

---

### Official Review · Reviewer_g2Pg · 2025-11-01

**Soundness:** 2
**Presentation:** 3
**Contribution:** 3
**Rating:** 6
**Confidence:** 3

**Summary:**

This paper introduces MTS-UNMixers, a novel model for multivariate time series forecasting (MTSF) designed to address the challenge of highly mixed temporal and channel features in high-dimensional data. Its core mechanism is "channel-time dual unmixing", which establishes an explicit mapping from historical to future sequences by decomposing the series into key "bases" and "coefficients". Architecturally, the model employs a standard Mamba network to capture temporal causality and estimate channel coefficients, while using a bidirectional Mamba to process non-causal bidirectional channel interactions and extract the shared time bases. Experimental results demonstrate that MTS-UNMixers significantly outperforms existing methods across multiple benchmark datasets.

**Strengths:**

1. The paper is generally well-organized and clearly structured.

2. The proposed method is novel and presents a unified approach to addressing challenges in both the channel and time dimensions within the MTSF domain, achieving strong experimental results.

3. The paper offers a novel perspective by reformulating the MTSF problem as a matrix decomposition task across the channel and time dimensions. This mechanism establishes an explicit mapping between historical and future sequences, a notable advancement in model interpretability over existing black-box approaches.

**Weaknesses:**

1. A primary motivation for the unmixing model is its physical interpretability (highlighted in the Abstract and Introduction). However, the paper provides no qualitative results to support this claim.

2. The method lacks strong theoretical justification. While temporal unmixing is a common practice, the paper’s direct extension of this formulation to the channel dimension (as in Eq. (6)) is presented by analogy and lacks a rigorous theoretical foundation for its validity in modeling non-causal channel correlations.

3. The proposed model relies heavily on Mamba blocks. However, the experiments omit comparisons with recent Mamba-based baseline models for MTSF, a significant gap in the experimental evaluation.

4. The notation for the dimensions of the input $\mathrm{X}$ is inconsistent and confusing throughout the paper. Section 2.1 defines $\mathrm{X} \in R^{T \times N}$, but the formulation for temporal mixing (e.g., Eq. (2)) implies $\mathrm{X} \in R^{N \times T}$.

**Questions:**

1. The paper's claim of interpretability needs to be substantiated. Could the authors provide visualizations or a qualitative analysis of the learned bases and coefficients to support this?

2. Sharing the temporal basis $A_t$ (e.g., seasonality) is reasonable, but sharing the channel coefficients $S_c$ (interpreted as static channel correlations) is a very strong assumption. It implies that inter-variable relationships remain constant over time, which seems unlikely for dynamic domains such as financial or traffic forecasting. How do the authors justify this static assumption, and wouldn’t it limit the model’s generalization capability?

3. Why did the authors choose to share $A_t$ and $S_c$, instead of the alternative (e.g., sharing $A_c$ and $S_t$)? The paper does not provide a justification, making this design choice appear arbitrary.

4. The experimental comparisons should be updated to include more recent and relevant baselines.

---

> ### Author Response · Authors · 2025-11-24
> **Response to Reviewer g2Pg (Part 1 / 2)**
>
> > Q1: On the validity and empirical verification of the physical interpretability & Q2 & Q3: On the choice and appropriateness of using $A_t$ and $S_c$  as shared structures
>
> ---
>
> We sincerely thank the reviewer for the careful reading and the insightful questions. Since our method follows a closed-loop logic of **hypothesis → modeling → verification**, the concerns raised in Q1 and Q2 are conceptually interconnected. We therefore address them together in this section to provide a clearer and more coherent explanation. The full theoretical details and extended empirical evidence have been incorporated into the revised manuscript, specifically in the Mixing Model section and the newly added Appendix B, and we kindly invite the reviewer to refer to these updated materials.
> ## **1. Theoretical Assumption**
>
> ### **1.1 Background of Physical Interpretability (Q1)**
> In our modeling framework, we assume that multivariate time series are composed by mixing a smaller set of more fundamental latent basis signals. Based on this assumption, the model also exhibits a dynamic–static separation property. The static invariants correspond to physical attributes, topological structures, and inherent response patterns that remain stable over time, while the dynamic variables reflect the instantaneous states of the system, external driving factors, and specific behavioral patterns that evolve over time. These considerations guide us to design the mixing model as follows:
>
> - **Temporal mixing (Channel-wise)**
>   $ X = A_c \in \mathbb{R}^{T \times k} \cdot S_c \in \mathbb{R}^{k \times N} $
>
> - **Channel mixing (Temporal-wise)**
>   $ X^\top = A_t \in \mathbb{R}^{N \times k} \cdot S_t \in \mathbb{R}^{k \times T} $
>
> ---
>
> ### **1.2 Reasonableness of Component Definition and Sharing (Q2 & Q3)**
>
> Based on the above hypothesis, the definition and partition of dynamic and static components are as follows.
>
> **1. Static invariants (shared components)**
> The latent channel structure ($A_t$) and the channel contribution coefficients ($S_c$) together form the system's time-invariant physical essence. This design compels the model to extract and share these core features, which is a theoretical necessity for achieving generalization.
>
> - **Channel basis structure ($A_t$)**
>   This represents stable latent structures that are formed jointly by multiple channels and persist across all time steps. It can be understood as "latent factors" or "groups of entities" (for example, groups of residential users, industrial subnetworks, or combinations of key sensors). These structures should correspond to one or more understandable common physical driving factors. For instance, in a typical economic system, financial factors, market sentiment, and supporting policies can all serve as common driving factors, where each column defines the portfolio weights of one factor. As another example, a common driving factor composed of all temperature-related variables (such as oil temperature and various loads) with positive weights can be physically interpreted as an "environmental temperature effect".
>
> - **Channel contributions ($S_c$)**
>   Quantifying the response strength of each physical variable in the temporal basis $A_c$ can clearly reveal synchronous, inverse, or lagging relationships among variables. For example, temperature variables exhibit strong responses to seasonal fluctuations.
>
> **2. Dynamic variables (non-shared components)**
> These include the shared temporal patterns ($A_c$, representing the concrete temporal patterns experienced by the system) and the temporal activations ($S_t$, describing the instantaneous states of the factors). Together, they form the "main channels of expression" for system dynamics. In order to flexibly fit the data, the model must keep these components non-shared so as to fully capture temporal evolution.
>
> - **Temporal basis patterns ($A_c$)**
>   This reflects the model's ability to abstract the macroscopic temporal behavior of the system. It extracts and separates macroscopic temporal patterns from the data that are aligned with known physical rhythms, such as seasonal variations (e.g., increased heating load in winter and increased cooling load in summer) and daily cycles (e.g., higher temperature during the day and lower temperature at night).
>
> - **Temporal activations ($S_t$)**
>   This represents the activation strength or influence of each latent channel structure at every time step. It is a time series that describes how each latent factor or group evolves over time. For example, for the "environmental temperature effect" structure, its temporal activation series should exhibit annual seasonal fluctuations consistent with actual temperature changes.

---

> ### Author Response · Authors · 2025-11-24
> **Response to Reviewer g2Pg (Part 2 / 2)**
>
> Theoretically, generalization stems from separating dynamic and static components. While traditional models overfit historical patterns, our approach enforces learning static representations ($A_t$ and $S_c$), reframing forecasting as inferring dynamic behaviors ($A_c$ and $S_t$) using known physical principles. This enables reasoning about unseen future patterns through essential features.
>
> Conversely, sharing dynamic components is physically implausible. Sharing $A_c$ assumes immutable temporal patterns (e.g., identical annual seasonality), while sharing $S_t$ presupposes constant factor activations (e.g., daily identical electricity patterns). Thus, sharing static components $A_t$ and $S_c$ remains the only logically coherent and physically meaningful design.
>
>
> ## **2. Unmixing Model Design (Q2 & Q3)**
>
> The elegance of our dual-path unmixing design lies in the fact that its output targets are inherently determined by the structure of its input data, rather than chosen arbitrarily. This perfectly forms a closed loop in our logic. The detailed process has been updated in the appendix.
>
> - **Temporal unmixing**: Since the input is the complete history of a single channel $X[:,j]$, the encoder analyzes its entire life cycle and ultimately outputs the static and inherent response weights of this channel to all shared temporal patterns $A_c$, i.e., $S_c$.
>
> - **Channel unmixing**: Since the input is an instantaneous snapshot of all channels, the encoder, by comparing a large number of system states at different time steps, ultimately learns a static and stable latent structure composed of channels that persists over time, i.e., $A_t$.
>
> Therefore, according to physical logic, the model we construct necessarily produces the shared static features $S_c$ and $A_t$.
>
> ## **3. Visualization-based Validation (Q1)**
>
> Visualization analysis using the ETTh1 and Electricity datasets has been provided and updated in **Appendix B** of the manuscript. We invite you to take a look!
>
> ---
> > Q4: Adding Mamba-related time series forecasting baselines
>
> Thank you for the suggestion. We fully agree on the importance of including Mamba-based models as baselines for time series forecasting. In response, we have added the following results to the main table of the revised manuscript:
>
> | Dataset  | TimeMachine[1] MSE | TimeMachine MAE | S-Mamba[2] MSE | S-Mamba MAE | SiMBA[3] MSE | SiMBA MAE | MTS-UNMixers MSE | MTS-UNMixers MAE |
> |----------|---------------------|-----------------|----------------|-------------|--------------|-----------|-------------------|------------------|
> | ETTh1    | 0.439               | 0.439           | 0.450          | 0.455       | 0.473        | 0.455     | **0.423**         | **0.417**        |
> | ETTh2    | 0.387               | 0.419           | 0.395          | 0.434       | 0.452        | 0.448     | **0.345**         | **0.379**        |
> | ETTm1    | 0.399               | 0.407           | 0.401          | 0.418       | 0.444        | 0.466     | **0.380**         | **0.389**        |
> | ETTm2    | 0.276           | 0.335           | 0.277          | 0.317       | 0.338        | 0.370     | **0.270**           | **0.316**        |
> | Weather  | 0.252               | 0.284           | 0.245          | 0.282       | 0.255        | 0.272     | **0.239**         | **0.265**        |
> | ECL      | 0.183               | 0.272           | 0.195          | 0.285       | 0.199        | 0.271     | **0.177**         | **0.268**        |
> | Traffic  | 0.425               | 0.307       | **0.423**      | **0.299**   | 0.513        | 0.422     | 0.466             | 0.293            |
>
> The experiments show that even compared with the latest Mamba-based MTSF models, MTS-UNMixers achieve the best performance on six out of seven datasets. This demonstrates the effectiveness of our approach, and suggests that the advantage comes from the proposed dual-path unmixing framework itself, rather than from the stacking of underlying encoders.
>
> [1] Timemachine: A time series is worth 4 mambas for long-term forecasting. arXiv
> [2] Is Mamba Effective for Time Series Forecasting? arXiv
> [3] SiMBA: Simplified Mamba Based Architecture for Vision and Multivariate Time series. arXiv
>
> ---
>
> > W4: Adding Mamba-related time series forecasting baselines
>
> Thank you very much for the detailed observation. We clarify that throughout this paper, the multivariate time series is consistently defined as $X \in \mathbb{R}^{T \times N}$, where $T$ denotes the time length and $N$ denotes the number of channels. This definition is never changed. Temporal mixing and Channel mixing take input tokens as $X[:, j]$ (a single-channel time series) and $X[t, :]$ (a channel snapshot at a single time point), respectively. They are simply slices of the same $X$ along different axes, rather than redefinitions of its dimensionality. We have revised the manuscript to remove any potential ambiguity.

---

### Author Response · Authors · 2025-11-24
**General Response and Summary of Revisions for All Reviewers**

Dear all reviewers,
We sincerely thank all reviewers for their careful evaluation of our manuscript and for the thoughtful and valuable comments provided. We are especially grateful for the recognition of our key innovations, including the unmixing framework and the shared mechanism, as well as for the constructive suggestions regarding theoretical rigor, experimental completeness and clarity of presentation.

Following the reviewers’ feedback, we have undertaken a thorough and substantial revision of the manuscript. All changes are highlighted in blue in the updated submission for ease of review. The main revisions and improvements are summarized as follows.
1. We reconstructed the mathematical formulation of the mixing and unmixing framework to ensure strict consistency between the two paths in both model structure and physical interpretation.
2. We clarified the role of the shared bases through a dynamic–static decoupling perspective, making their mechanism in capturing cross-channel stable structures more explicit.
3. We added a systematic closed-loop visual analysis that directly illustrates how the model performs the inverse operations of the mixing process through the two unmixing paths, thereby validating the core mechanism.
4. We expanded the experimental comparisons to include Mamba-based methods and recent state-of-the-art models, providing comprehensive evidence of the robustness and stability of our unmixing structure across various data scales and noise conditions.

We acknowledge that the response and newly added materials are extensive and may impose additional workload for re-review. We sincerely apologize for this and are deeply grateful for your time and effort.

Once again, we thank all reviewers for their valuable input. We look forward to further improving this work based on your guidance. Please feel free to let us know if any additional clarification is needed.

---

### Note · Program_Chairs · 2026-01-17
**Submission Desk Rejected by Program Chairs**

The following references in this submission do not refer to real documents and/or have major errors in bibliographic information:

 Chao Chen, Andre Petros, et al. Freeway traffic prediction using machine learning algorithms. Journal of Transportation Engineering, 127(1):14-22, 2001.